# Optimal searching behaviour generated intrinsically by the central pattern generator for locomotion

David W Sims[1,2,3†]*, Nicolas E Humphries[1†], Nan Hu[4], Violeta Medan[5,6], Jimena Berni[4†]*

[1]The Marine Biological Association of the United Kingdom, Plymouth, United Kingdom; [2]Ocean and Earth Science, National Oceanography Centre Southampton, University of Southampton, Southampton, United Kingdom; [3]Centre for Biological Sciences, University of Southampton, Southampton, United Kingdom; [4]Department of Zoology, University of Cambridge, Cambridge, United Kingdom; [5]Departamento de Fisiología, Biología Molecular y Celular, Facultad de Ciencias Exactas y Naturales, Universidad de Buenos Aires, Ciudad Universitaria, Buenos Aires, Argentina; [6]Instituto de Fisiología, Biología Molecular y Neurociencias (IFIBYNE-UBA-CONICET), Buenos Aires, Argentina

**Abstract** Efficient searching for resources such as food by animals is key to their survival. It has been proposed that diverse animals from insects to sharks and humans adopt searching patterns that resemble a simple Lévy random walk, which is theoretically optimal for 'blind foragers' to locate sparse, patchy resources. To test if such patterns are generated intrinsically, or arise via environmental interactions, we tracked free-moving *Drosophila* larvae with (and without) blocked synaptic activity in the brain, suboesophageal ganglion (SOG) and sensory neurons. In brain-blocked larvae, we found that extended substrate exploration emerges as multi-scale movement paths similar to truncated Lévy walks. Strikingly, power-law exponents of brain/SOG/sensory-blocked larvae averaged 1.96, close to a theoretical optimum ($\mu \cong 2.0$) for locating sparse resources. Thus, efficient spatial exploration can emerge from autonomous patterns in neural activity. Our results provide the strongest evidence so far for the intrinsic generation of Lévy-like movement patterns.

*For correspondence:
dws@mba.ac.uk (DWS);
jb672@cam.ac.uk (JB)

†These authors contributed equally to this work

Competing interests: The authors declare that no competing interests exist.

## Introduction

The generation of rhythmic patterns of movement is a key feature of behaviours that are conserved across the animal kingdom (*Ijspeert, 2008*). Central pattern generators (CPGs) are motor neural circuits capable of generating rhythmic activity when activated that require no sensory or descending inputs of phasic timing information (*Ijspeert, 2008*; *Marder et al., 2005*; *Berni et al., 2012*; *Berni, 2015*). However, complex sequences of behaviour entirely organised within the nervous system without the need for sensory inputs from environmental influences or sensory feedback during its course are seemingly rare in nature (*Marder et al., 2005*; *Slater, 1999*). For instance, finding resources requires efficient movement through complex environments and is an ability key to animal survival (*Ijspeert, 2008*; *Sims et al., 2008*). Foraging, in which an animal searches widely to obtain food resources, is one such complex sequence of behaviour, comprising physiological, sensory and cognitive processes modified in response to environmental fluctuations. Apparently complex patterns of behaviour can potentially arise from simple movement 'rules' (*Sims et al., 2008*; *Maye et al., 2007*; *Wearmouth et al., 2014*), however it remains unknown whether autonomous movements resulting in spatial exploration could result in optimal searches.

Autonomous generation of movement patterns that are optimal for finding resources should be advantageous in naïve and inexperienced animals. It has been proposed (*Maye et al., 2007*; *Wearmouth et al., 2014*; *Kölzsch et al., 2015*; *Humphries et al., 2012*; *de Jager et al., 2014*; *Sims, 2015*; *Reynolds, 2015*) that foraging patterns described by apparent Lévy random walks in freely moving animals may represent intrinsically generated behaviour that is exhibited when a searcher has incomplete information about resource location, such as when there is an absence of external clues or cues, or no information about the environment provided by memory (*Sims et al., 2008*). Therefore, where there is a lack of knowledge about resource location, autonomously generated search programs based on Lévy walks may represent efficient patterns to adopt until clues and cues are encountered by moving.

A Lévy random-walk search pattern can be generated by drawing displacements (move steps) from a probability distribution with a heavy power-law tail, such that $P(l) \sim l^{-\mu}$, with $1 < \mu \leq 3$ where $l$ is the move step length between turns and $\mu$ the power law exponent (*Figure 1A,B*). A Lévy walk iterated over many time intervals will be distributed much further from its starting position than a Brownian walk of the same length because small-step 'walk clusters' are interspersed by long 'steps' to new locations, with this pattern repeating across all scales resulting in a fractal pattern of walk clusters with no characteristic scale (*Sims et al., 2014*; *Reynolds, 2018*). Theory predicts that Lévy walk search strategies are optimal where resources are sparse and distributed unpredictably (*Viswanathan et al., 1999*; *Viswanathan et al., 2011*; *Humphries and Sims, 2014*; *Wosniack et al., 2015*) – such as in uncertain or dynamic environments where the spatial scaling patterns of search behaviour cannot be tuned to target distributions (*Reynolds, 2018*) – whereas Brownian (exponential) walks are sufficiently efficient when resources are highly abundant (*Viswanathan et al., 1999*; *Viswanathan et al., 2011*; *Humphries and Sims, 2014*; *Wosniack et al., 2015*). However, a Lévy walk with an inverse square power-law distribution of move steps ($\mu \sim 2$) is optimal across a very broad range of resource distributions and abundances (*Humphries and Sims, 2014*; *Wosniack et al., 2015*). Because Lévy walks can optimise search efficiencies in this way, it has been proposed that natural selection should have led to adaptations for Lévy walk foraging [the Lévy foraging (LF) hypothesis] (*Sims et al., 2008*; *Viswanathan et al., 1999*; *Viswanathan et al., 2011*; *Viswanathan et al., 2008*). Movement patterns resembling Lévy walks have been observed in diverse organisms (*Sims et al., 2008*; *Maye et al., 2007*; *Kölzsch et al., 2015*; *Humphries et al., 2012*; *Cole, 1995*; *Humphries et al., 2010*; *de Jager et al., 2011*; *Bazazi et al., 2012*; *Hays et al., 2012*; *Raichlen et al., 2014*), such as jellyfish, molluscs, insects, fish, birds and mammals including human hunter-gatherers, and even in fossil trails of extinct invertebrates (*Sims et al., 2014*), leading to the idea that search strategies naturally evolved to exploit optimal Lévy walks (*Sims et al., 2008*; *Viswanathan et al., 2008*). However, the origin of apparent Lévy-like behaviour is unclear (*Sims, 2015*; *Sims et al., 2014*) and remains contentious (*Reynolds, 2015*; *Reynolds, 2018*; *Viswanathan et al., 2011*; *Benhamou, 2007*), not least because a candidate generative mechanism for Lévy movement has not been identified unequivocally (*Kölzsch et al., 2015*; *de Jager et al., 2014*; *Reynolds, 2015*; *Sims et al., 2014*).

Although Lévy-like patterns appear prevalent in organism movement patterns it has been argued that Lévy walks lack a biological basis or are too simplistic to explain the complexity and diversity of animal movements (*Reynolds, 2015*; *Reynolds, 2018*; *Benhamou, 2007*). While it is not expected that Lévy-like movement will be temporally ubiquitous in animal behaviour, since it is only advantageous when searching with incomplete information (*Sims et al., 2008*; *Viswanathan et al., 1999*; *Humphries and Sims, 2014*), recent studies have hypothesised two principal mechanisms to account for Lévy movement patterns (*Sims, 2015*): it arises from (i) endogenous neurophysiological processes (*Maye et al., 2007*; *Wearmouth et al., 2014*; *Kölzsch et al., 2015*; *Humphries et al., 2012*; *de Jager et al., 2014*; *Sims et al., 2014*) resulting in behavioural adaptations to different resource distributions (*Sims et al., 2008*; *Humphries et al., 2010*) – the intrinsic hypothesis; or that (ii) sensory interactions of animals moving along straighter, ballistic type paths in fractal environments, for example, power-law distributions of resource patches, give rise to Lévy patterns as an emergent phenomena (*Reynolds, 2015*; *Reynolds, 2018*; *Boyer et al., 2006*) – the extrinsic hypothesis. Nevertheless, there have been no studies to date that have unequivocally tested both hypotheses in the same model organism.

Our ability to specifically and acutely manipulate neuronal activity in *Drosophila* larvae during a spatial exploration sequence makes it a suitable model for testing the Lévy foraging hypothesis and

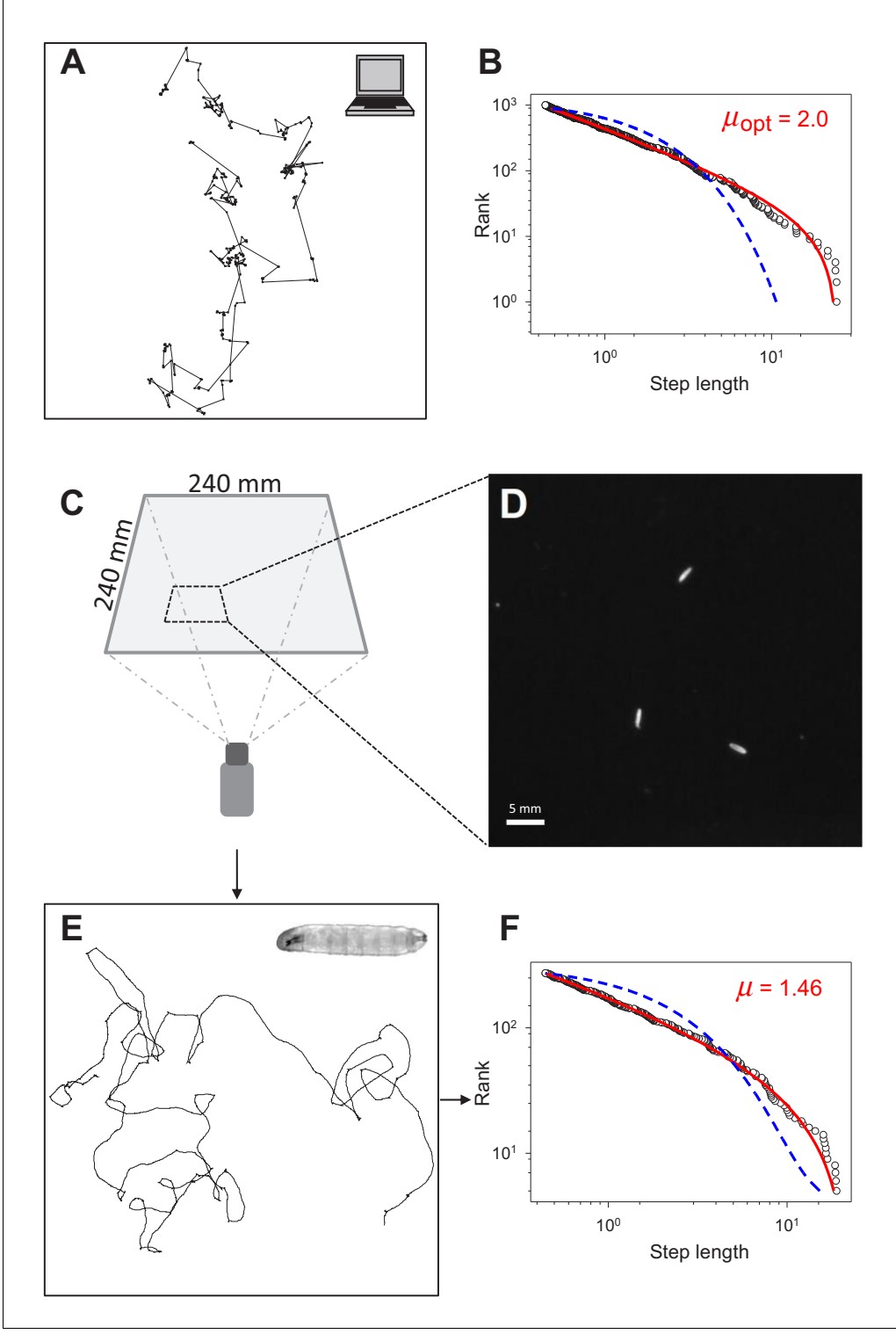

**Figure 1.** Testing for Lévy-like search patterns in *Drosophila* larva tracks. (A) Computer simulated path with Lévy-distributed move step lengths (truncated power-law exponent, μ = 2.0) between turning points. (B) Rank step-length-frequency distribution of steps in (A) with Maximum Likelihood Estimation (MLE) fits of truncated power law (red line) and exponential (blue) model distributions. Recovered exponent (μ) of the truncated power law fit shown is 1.996. (C) Set-up and dimensions of the temperature-controlled, dark experimental arena (see Materials and methods) used to (D) record the positions of larvae to determine (E) the movement trajectories of individual larva (inset), prior to (F) path discretization and MLE model fitting and selection analysis.

to explore its mechanisms. Here, we used two transgenic lines of flies: BL (for brain lobes) in which the brain and part of the SOG function can be acutely and reversibly impaired in free-moving larvae, and BLsens (for BL and sensory), which allows us to block the brain lobes, the SOG and all somato-sensation (*Berni et al., 2012*; *Berni, 2015*). Movements in these larvae comprise sequences of forward peristaltic contractions (crawls) interspersed with pauses and reorientations (turns) in which the brain and part of the SOG are not required for normal performance of an exploratory routine; patterns are produced by autonomous activity of the thoracic and abdominal nervous system (*Berni et al., 2012*; *Berni, 2015*). Because the BL line may not be expressed in all neurons, we performed electrophysiology experiments to evaluate the strength of neuronal activity blockade. We also tested a transgenic line specific for the mushroom body, MB247 (*Pauls et al., 2010*), to determine the potential contribution of cholinergic neurons from this structure of the brain in free-moving larvae. Finally, to confirm the results obtained affecting activity in the brain lobes, SOG and sensory system, we induced neuronal death expressing the proapoptotic genes (rpr and hid) with the BLsens line.

We used this model system to test whether extended substrate exploration of *Drosophila* larvae is an intrinsic, spontaneous pattern resembling a Lévy search in the absence of exteroception and proprioception.

## Results

We recorded movement paths of individual larva using high-resolution video tracking extending for 1 hr (*Figure 1C–E*) (Materials and methods). A Kalman filter used for track smoothing to reduce effects of video tracking errors combined with computation of track step lengths reconstructed the larva movement paths reliably (*Figure 2A*; see Materials and methods). Across all controls and treatments in this study 658 larva tracks were recorded; 187 (28.4 %) did not meet the criteria required for reliable track analysis to support testing of the Lévy foraging hypothesis (*Humphries et al., 2013*) and were discarded prior to track modelling analysis (*Supplementary file 1*). The criteria for discarding individual larva tracks were: (i) tracks with fewer than 200 move steps (126 tracks), or (ii) tracks where the larva interacted with the arena edge (e.g. travelled along it; 52 tracks), or (iii) where no computed steps were longer than the 0.44 mm minimum resolution for a step length (nine tracks) (see Materials and methods) (*Humphries et al., 2010*). *Figure 2A* and *Figure 2—figure supplement 1* provides illustrations of the track processing for an example track with the detailed analysis of the turning behaviour. Turns were identified as the reversal in direction in a 1D projection of the 2D movement patterns. This method is unbiased and preserves the distribution properties of the original 2D trajectory (*Humphries et al., 2013*) (see Materials and methods). We undertook sensitivity analyses to examine how variations in Kalman filter parameters and in the minimum step resolution value affected truncated power-law fitting and exponents (μ). We found no significant effects of different Kalman filter parameters or minimum resolution step values on mean μ values (*Supplementary file 2*), demonstrating that the μ values were robust to relatively broad alterations in video tracking resolution and track smoothing (*Figure 2—figure supplement 2*). Similarly, there were no significant differences between movement patterns as measured by μ values of larva before and after collisions with other larva or the arena edge within BL/+ and *shi^{ts}/+* treatments (*Figure 2—figure supplement 3*) – indicating stationarity in movement data statistics (for additional stationarity tests see 'Exploration strategy in an environment with minimal external cues' section) – thus, track sections occurring after brief collisions were retained for analysis.

For the remaining 471 tracks across all controls and treatments we tested for Lévy walks by fitting truncated power law (Pareto-Lévy) and exponential distributions to empirical move-step length frequency distributions using Maximum Likelihood Estimation (MLE) for parameter fitting and Akaike Information Criteria weights (*w*AIC) for model selection (illustrated in *Figure 1E,F*; Materials and methods) (*Humphries et al., 2013*). We tested for truncated power laws rather than pure power laws because animal movements are naturally bounded, even within the relatively large experimental arena used (*Figure 1C*). As well as truncated power laws and exponential models we also tested power-law, truncated exponential, log-normal (a heavy-tailed distribution), and gamma distributions in addition to composite Brownian walk distributions comprising proportions of two, three, and four exponential distributions (*Sims et al., 2014*).

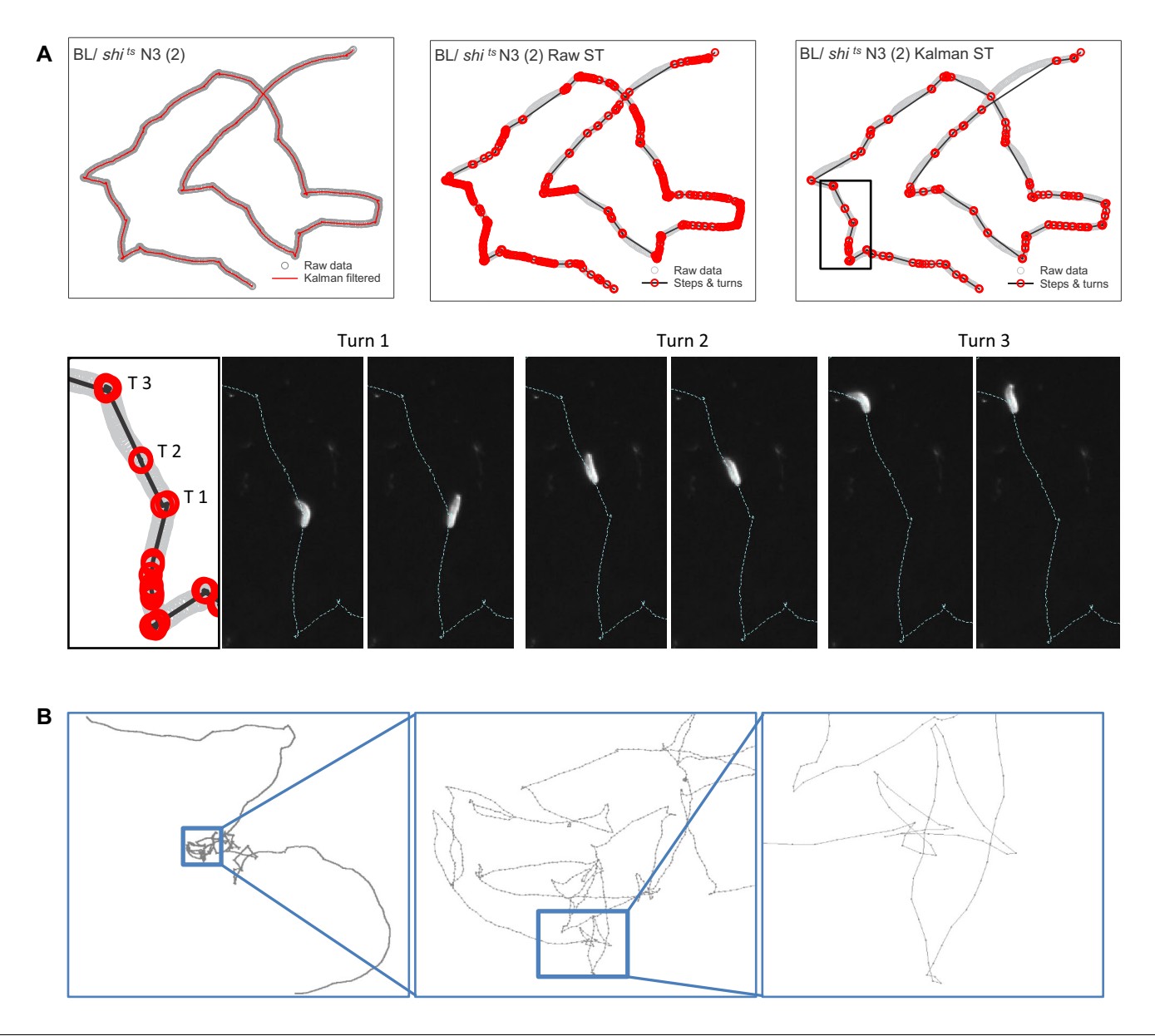

**Figure 2.** Estimating move step lengths in *Drosophila* larva tracks across broad scales. (A) Examples of track processing and turn identification. Row 1, column one shows an example of the effect of the Kalman filter on raw track data; column two shows the steps and turns that would result if raw track data were analysed; column three shows the steps and turns identified following the Kalman smoothing of the raw data. Row two shows the turns executed by the larva for a short section of the track. Note that the method detects small (turn 2, (T2) and large turns (turn 1 and 3 (T1, T3)). (B) Example of normal substrate exploration (control BL / + larva) shows a similar pattern of complexity at all scales, characteristic of scale-invariant Lévy walk patterns.

The online version of this article includes the following figure supplement(s) for figure 2:

**Figure supplement 1.** Analysis of curved paths.

**Figure supplement 2.** Effects of Kalman filter parameter changes on an example larva track.

**Figure supplement 3.** Effect of edge collision on search strategy.

## Exploration strategy in an environment with minimal external cues

At the control temperature of 22°C where normal movements and feeding responses were expected (*Hoffmann, 2010*), we found for control larva experiencing minimal external stimuli, e.g. visual, olfactory and gustatory, that 30 trajectories (70 %) of BL/+ (*elav*-Gal4, *tsh*-Gal80/+,+; *cha3.3*-Gal80/

+) larvae with intact nervous systems, and 34 (79 %) of *shi*^ts/+ (UAS-*shi*^ts/+) larvae with intact nervous systems and inactivated shibire^ts, were best fitted by truncated Lévy power-law distributions (*Supplementary file 1*; Figure 5). Only 14 tracks at 22°C (16%) were best fit by an exponential distribution, while eight tracks (9%) were best fit by neither model (*Supplementary file 1*).

For the BL/+ larva used, despite 33°C being above the described preferred temperature of the larvae, we found that it was still within a range where feeding behaviour was not affected (*Figure 3B*). This suggests that movements of BL/+ at the restrictive temperature reflected normal searching movements to locate resources. Similarly, in control larva at 33°C to determine the effects on movement paths of the restrictive temperature (33°C) at which UAS-*shi*^ts blocks neuronal transmission (*Figures 3* and *4*), we found 57% of BL/+ larvae and 77% of *shi*^ts/+ larvae move-step distributions were best fitted by truncated Lévy power-law distributions (*Figure 5*), with 14 tracks (13%) best fit by the exponential model; 23 tracks (21%) were best fit by neither model) (*Supplementary file 1* and *3*). There were no fits to any other model distributions tested, except for seven tracks that were best fit by pure power law distributions (*Supplementary file 4* and *6*).

Movements comprised spatially intensive clusters of shorter move steps with frequent turns interspersed with more extensive, straighter steps of varying length (*Figure 2B* and *5*). This intensive/extensive movement pattern was evident at increasing scales within the test arena (*Figure 2B*) and is a characteristic of scale-free Lévy patterns (*Kölzsch et al., 2015*; *Viswanathan et al., 1999*). We confirmed this observation using tortuosity analysis of the control tracks at both temperatures (*Table 1*; Materials and methods). MLE analysis of the frequency distributions of straightness index 'jumps', describing the transition from spatially intensive to extensive movement clusters across different scales (cluster sizes) within each individual time-series, demonstrated good fits to truncated power-laws (127 of 136 tracks; 93%) averaging 1.98 orders of magnitude of the data (*Table 1*), which are characteristics that are expected for a Lévy pattern (*Kölzsch et al., 2015*).

At the environmentally relevant control temperature (22°C), model fitting of move step length frequency distributions showed that all 31 BL/+ and 34 *shi*^ts/+ tracks fitting truncated power-laws showed characteristics that are consistent with Lévy random walks. Firstly, we found BL/+ and *shi*^ts/+ move-step distributions to have power-law exponents ($\mu$) within the Lévy range ($1 < \mu \leq 3$) (BL/+: mean $\mu$, 1.47 ± 0.36 S.D., $n$ = 30; *shi*^ts/+: mean $\mu$, 1.49 ± 0.35 S.D., $n$ = 34) (*Table 2*; *Supplementary file 3*). Cumulative probability distribution (CPD) analysis confirmed that for the sample exponent parameters it was predicted that 93.2% and 98.4% of BL/+ and *shi*^ts/+ $\mu$ exponents within populations, respectively, were estimated to fall within the Lévy range (*Table 3*). In addition, we confirmed stationarity in the movement pattern data by finding no differences in mean $\mu$

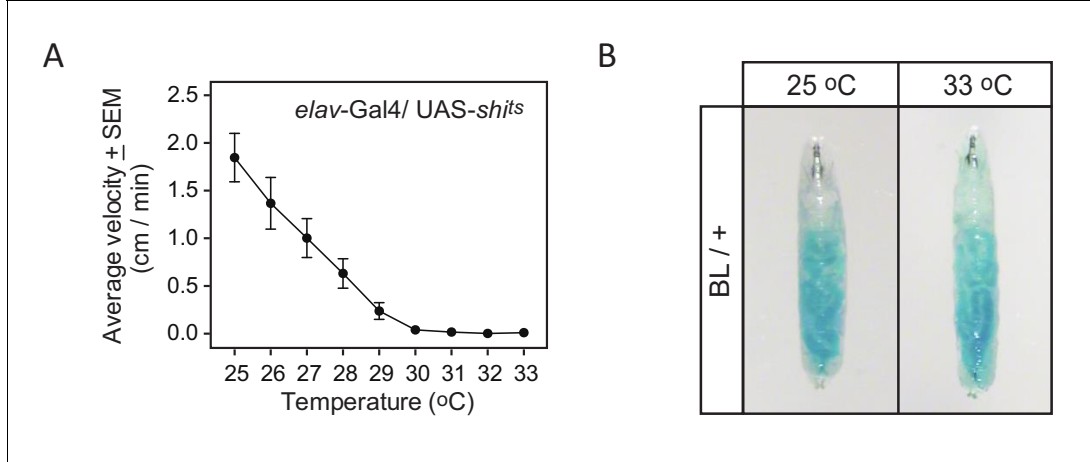

**Figure 3.** Effect of temperature on activity and feeding. (**A**) The curve of inactivation of synaptic transmission with increasing temperature demonstrated by decrease in the average velocity of *elav*-Gal4/UAS *shi*^ts third instar larvae as a function of temperature. Ten larvae were placed on the arena and their behaviour was evaluated as the temperature increased. At 30°C and above the larvae are paralysed indicating that synaptic transmission is blocked panneurally. (**B**) Effect of temperature on feeding behaviour. Larvae were placed on small patches of yeast (50% dry yeast in water with 0.1% brilliant blue) located on the arena for 30 min. A blue gut indicates that larvae have been feeding similarly at 25°C and 33°C. This shows that even if 33°C is above the described preferred temperature of the larvae (see Materials and methods), it is still within a range where feeding behaviour is not affected.

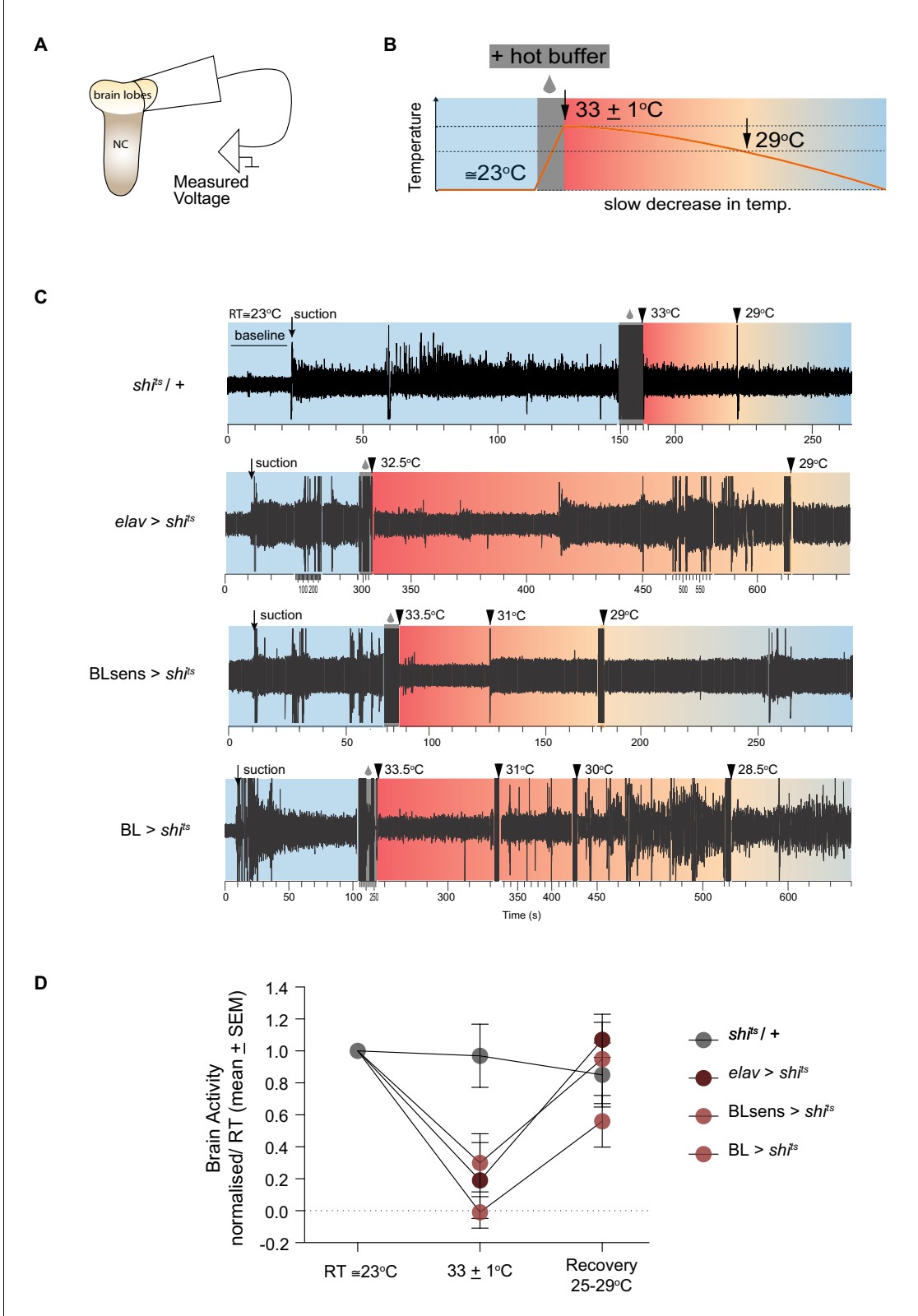

**Figure 4.** Shibire[ts] mediated inhibition of brain activity. (A–B) Schematic of the experimental design for extracellular recordings. (A) Early third instar larvae were dissected and a suction electrode was attached to one brain lobe. NC denotes the nerve cord (thoracic and abdominal segments). (B) Neuronal activity was measured at reference temperature (RT ≅23°C) shaded in light blue and during the hot temperature treatment (shaded red), which started with hot buffer superfusion (grey box) until the restrictive temperature was reached (33 ± 1°C) and continued until ≅29°C (recovery) where

*Figure 4 continued on next page*

*Figure 4 continued*

normal levels of activity were reached in *shi$^{ts}$* expressing samples. (**C**) Representative traces of measured activity for each genotype. Baseline until suction and recording at RT are shaded light blue. The suction produced a mechanical artifact followed by an increase in the recording amplitude. Hot buffer superfusion and temperature probe (grey box) also produced an artifact. The increase in temperature did not change brain activity in *shi$^{ts}$/+* lobes. However, there was a sharp decrease for *elav >shi$^{ts}$*, BLsens > *shi$^{ts}$* and BL >*shi$^{ts}$*. Brain activity gradually recovered until reaching levels that did not differ from RT activity. Note that the time scale has been adjusted to account for differences in recording period. (**D**) Normalised neuronal activity. Brain lobe response to temperature changes was quantified as the ratio between response recorded (minus the baseline) at (33 ± 1°C) and the RT. At restrictive temperature (33 ± 1°C), neuronal activity did not change in the negative control *shi$^{ts}$ /+* animals (n = 6), while it was blocked in the positive control *elav >shi$^{ts}$* (n = 5), and in the experimental treatments BL >*shi$^{ts}$* (n = 7) and BLsens > *shi$^{ts}$* (n = 5). A two way ANOVA for repeated measures showed differences within the temperature ($F_{(2, 57)}$=13.84; p<0.0001) and the genotype groups ($F_{(3, 57)}$=2.785; p=0.05) but no significant interaction. A post hoc Tukey's test comparing genotypes at each temperature showed significant differences only at (33 ± 1°C) between the control *shi$^{ts}$ /+* and the treatments, *elav >shi$^{ts}$*, BLsens > *shi$^{ts}$* and BL >*shi$^{ts}$* (p<0.0096, p<0.0007 and p<0.016, respectively).

values from the first and second halves of larva tracks each fitted by a truncated power-law (*Supplementary file 5* for results for all treatments). Second, the mean squared displacement (MSD) of a 2D trajectory describes the spatial extent of random motion where the scaling behaviour indicates short ($\alpha$ = 1) and long-range ($\alpha$ >1) correlations in the random walks. Hence, $\alpha$ = 1 describes normal Brownian random walks whereas $\alpha$ >1 characterises scale-free Lévy random walks (*Sims et al., 2008*; *Viswanathan et al., 2011*; *Bartumeus et al., 2005*). Therefore, we computed the MSD for each track and found that the means of $\alpha$ for individual BL/+ and *shi$^{ts}$/+* larvae were 1.34 and 1.26, respectively, which supports the presence of long-range correlations in the control larvae paths (*Table 4*). Lastly, the best fit truncated power-law model for each individual larva move-step distribution was greater than 1.5 orders of magnitude of the data (BL/+ mean, 1.63 ± 0.41 S.D., n = 30; *shi$^{ts}$/+* mean, 1.52 ± 0.32 S.D., n = 34) (*Table 2*; *Figure 6*), with some individual tracks spanning two orders of magnitude which is considered a rule of thumb for a candidate power law (*Stumpf and Porter, 2012*). These results demonstrate that when moving in environments with ambient temperatures similar to those found in *Drosophila* natural habitats, control BL/+ and *shi$^{ts}$/+* larvae exhibit movement patterns with Lévy-like characteristics.

For control treatments BL/+, *shi$^{ts}$/+* and MB247/+ (MB247-Gal4/+) larvae at the restrictive temperature (33°C) similar results were found. In summary, frequency distributions of straightness index cluster size and move step length distributions followed truncated power-law distributions over several orders (*Table 1*; *Supplementary file 3*), with mean µ exponents between 1.35 and 1.74 (*Table 2*; *Figure 5*) and cumulative probability distributions showing >90% of larvae trajectories were within the Lévy range (*Table 3*). Finally, MSD $\alpha$ values were between 1.37 and 1.41 confirming long-range correlations (*Table 4*). Therefore, BL/+, *shi$^{ts}$/+* and MB247/+ larvae (without synaptic blockade) moving for up to 1 hr time periods in larger environments with minimal exposure to external cues at the higher (restrictive) temperature of 33°C, exhibited movement patterns with Lévy-like characteristics that were similar to those of larvae at 22°C.

Furthermore, we also found that the turn angle distribution for all individual larva paths that were best fitted by truncated power-laws showed a near uniform distribution of large turn angles (50 and 175°, left or right) (*Figure 5—figure supplement 1*), a characteristic consistent with the uniform random turn-angle distribution (i.e. where all turn angles have equal probability) expected in a pure, idealised Lévy random walk (*Viswanathan et al., 1999*; *Bartumeus et al., 2005*). The presence of some directional persistence in our larva tracks (higher frequency of turn angles around 0°) indicates some short-range angle correlations. However, it has been widely demonstrated that the efficiencies of random Lévy walk searches remain robust to the presence of short-range correlations in directional persistence (*Bartumeus et al., 2005*; *Viswanathan et al., 2001*; *da Luz et al., 2001*; *Raposo et al., 2003*). This, taken together with the presence of truncated power-law (Lévy) fits to move step-length and walk-cluster size distributions, and the presence of long-range correlations in movement paths confirms the appropriateness of applying the Lévy random walk model to *Drosophila* larva tracks.

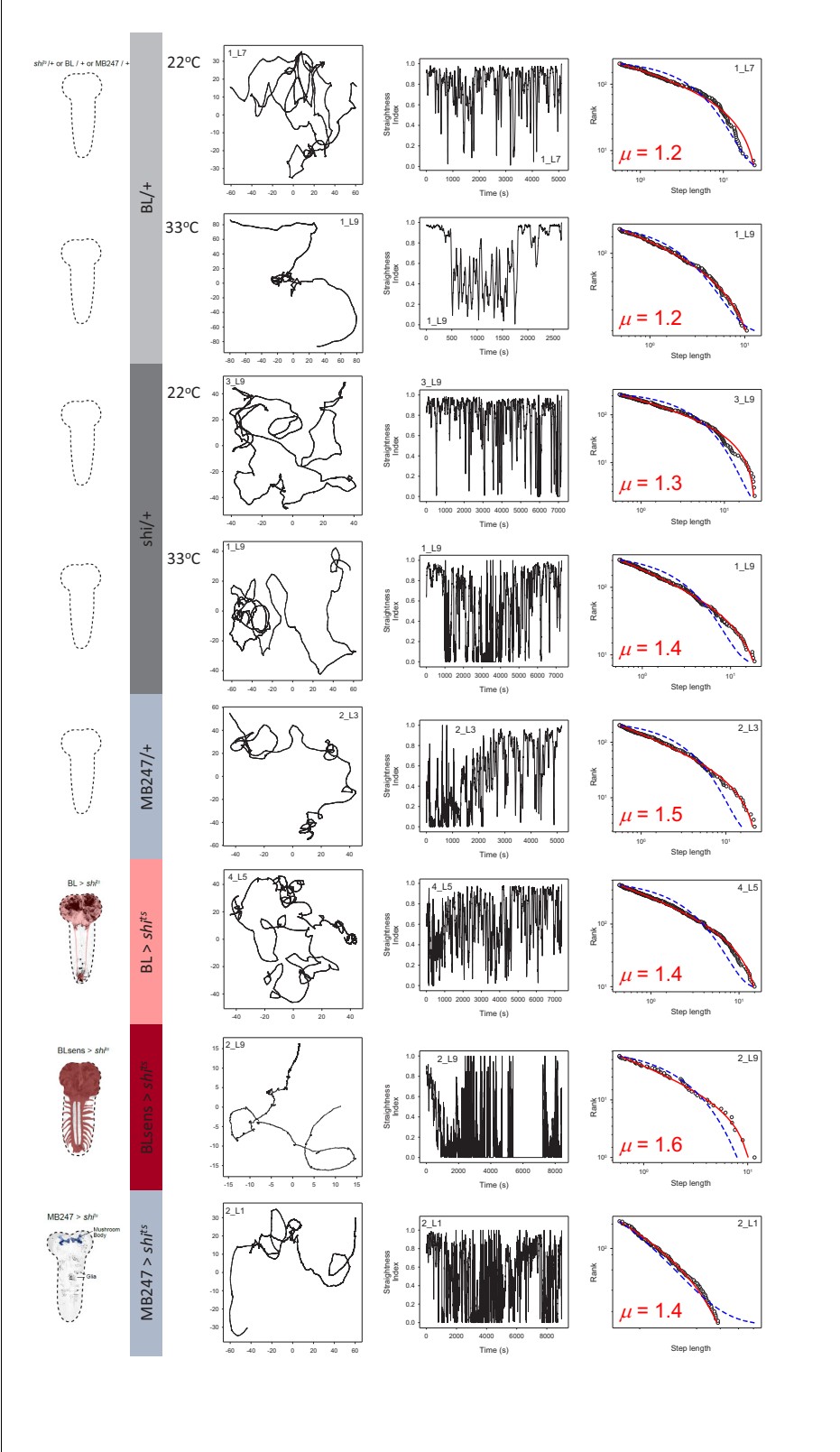

**Figure 5.** Control and brain blocked larvae movements. Example of larva trajectories for each of 8 experimental treatments. Each treatment is signified by the *Drosophila* brain outlines and colour coding on the left of panels (A). BL/+ (light grey) and *shi^{ts}*/+ (dark grey) treatments were conducted at normal (22°C) and restrictive temperatures (33°C) whereas MB247/+ and MB247 >*shi^{ts}* (light blue), and BL >*shi^{ts}* (pink) and BLsens > *shi^{ts}* (red) were tested at 33°C. (B) For each treatment an example larva trajectory is given with paths showing complex patterns of movement. All trajectories showed

*Figure 5 continued on next page*

*Figure 5 continued*

similar complexity across scales but which are not possible to illustrate in a single panel as here, but which are evident from path tortuosity analysis. Scale of paths are denoted by the *x,y* coordinates in mm. (**C**) Tortuosity analysis shows complex alternations of long, straighter move steps (higher straightness index, S.I., values) with short, spatially intensive clusters of move steps of different size (lower values). Note the BL >*shi*^ts larva with no function in the brain and a section of the suboesophageal ganglion (SOG), and BLsens > *shi*^ts larva with blocked brain lobes, SOG and all somatosensation, both show complex exploration movements similar to controls. (**D**) MLE model fitting and selection demonstrates larvae move-step length rank frequency distributions best fit by truncated Lévy power laws for each experimental treatment (fitted red line) compared to a Brownian walk (exponential model; blue line). Move step length frequency distributions were well approximated by a truncated Lévy power law across all treatments, with exponents ranging from 1.3 to 2.0. It is striking that BLsens > *shi*^ts larvae exponents were close to the theoretical optimum (µ ~2.0) indicating Lévy-like movements must be generated intrinsically within thoracic and abdominal neural circuitry.

The online version of this article includes the following figure supplement(s) for figure 5:

**Figure supplement 1.** The distributions of turning angles along larva paths.

## Lévy-like movement patterns are generated when brain processing is impaired

The demonstration of Lévy-like movement patterns in control larvae in an environment with minimal external stimuli is not conclusive support for an intrinsic mechanism underlying Lévy-like behaviour. *Figure 2A* shows the effect on movement of larva with panneural expression of a dominant negative temperature-sensitive form of Shibire^ts that blocks synaptic transmission (but does not affect electrical synapses) by depleting rapidly and almost completely the synaptic vesicles at the restrictive temperature of 33°C (*Koenig et al., 1983*; *Ohyama et al., 2013*). To test whether Lévy-like exploration movements are intrinsic to the neural circuitry of larvae in the absence of environmental sensory processing by the brain, we tracked individual BL >*shi*^ts (>stands for Gal4/UAS) larva with reduced synaptic activity in the brain and part of the suboesophageal ganglion (SOG) (*Berni et al., 2012*; *Berni, 2015*) (Materials and methods). Extracellular recordings of neuronal activity confirm that above 33°C, BL >*shi*^ts brain activity is impaired in a manner indistinguishable from the positive control *elav* >*shi*^ts, such that the proportion of activity left at 32°C compared to activity at 23°C was 0.30 ± 0.48 (S.D.) and 0.19 ± 0.53 (S.D.), respectively (Tukey *post hoc* test: BL >*shi*^ts vs *elav* >*shi*^ts: p=0.91) (*Figure 4*).

In 1 hr long trials of 77 tracks, 30% were discarded from further analysis (see criterion above; *Supplementary file 1*). However, 54 BL >*shi*^ts larva paths were sufficiently extensive to support reliable parameter estimation (*Humphries et al., 2013*). The BL >*shi*^ts larvae explored the arena as expected (*Berni et al., 2012*; *Berni, 2015*) (*Figure 5*), confirming here that movement paths were produced from peristaltic contractions and reorientations (turns) generated within the thoracic and abdominal circuitry alone. It was interesting that the movement patterns showed spatially intensive

**Table 1.** Tortuosity (cluster) analysis MLE results.

Summary of mean µ exponents and orders of magnitude of the data range fitted by truncated power-law model fits from tortuosity analysis. T (°C) denotes different environmental temperature of treatment.

| T (°C) | Treatment | No. ofTracks | Mean µ | SD | Orders of Magnitude | SD |
|---|---|---|---|---|---|---|
| 22 | BL/+ | 29 | 1.70 | 0.34 | 1.98 | 0.41 |
| 33 | BL/+ | 31 | 1.54 | 0.38 | 1.99 | 0.62 |
| 22 | *shi*^ts /+ | 33 | 1.85 | 0.44 | 1.89 | 0.39 |
| 33 | *shi*^ts /+ | 34 | 1.61 | 0.40 | 2.04 | 0.40 |
| 33 | MB247/+ | 21 | 1.39 | 0.27 | 1.87 | 0.45 |
| 33 | BLsens > *shi*^ts | 6 | 1.75 | 0.65 | 1.37 | 0.51 |
| 33 | BL > *shi*^ts | 23 | 1.67 | 0.40 | 1.61 | 0.61 |
| 33 | MB247 > *shi*^ts | 22 | 1.35 | 0.18 | 1.93 | 0.34 |
| 22 | New Blsens > *rpr,hid*_control | 61 | 1.74 | 0.33 | 2.11 | 0.37 |
| 32 | New BLsens > *rpr, hid* | 22 | 1.41 | 0.29 | 1.55 | 0.26 |

**Table 2.** MLE Analysis summary.

Summary of mean $\mu$ exponents and orders of magnitude of the data range fitted by truncated power-law model fits across experimental treatments. T (℃) denotes different environmental temperature of treatment. Means and standard deviations (SD) are given for exponents and data ranges of individual larva move-step length frequency distribution fits.

| T (℃) | Treatment | N tracks | Mu | SD | Orders of magnitude of the data | SD |
|---|---|---|---|---|---|---|
| 22 | BL/+ | 30 | 1.47 | 0.36 | 1.63 | 0.42 |
| 33 | BL/+ | 33 | 1.35 | 0.32 | 1.48 | 0.49 |
| 22 | $shi^{ts}$/+ | 34 | 1.49 | 0.35 | 1.52 | 0.32 |
| 33 | $shi^{ts}$ /+ | 39 | 1.54 | 0.43 | 1.52 | 0.38 |
| 33 | MB247/+ | 23 | 1.74 | 0.48 | 1.33 | 0.44 |
| 33 | BL > $shi^{ts}$ | 26 | 1.66 | 0.50 | 1.15 | 0.40 |
| 33 | BLsens > $shi^{ts}$ | 11 | 1.96 | 0.39 | 1.20 | 0.23 |
| 33 | MB247 > $shi^{ts}$ | 23 | 1.56 | 0.35 | 1.39 | 0.46 |
| 22 | BLsens > rpr, hid_control | 62 | 1.51 | 0.38 | 1.71 | 0.40 |
| 32 | BLsens > rpr, hid | 23 | 2.14 | 0.47 | 0.93 | 0.52 |

walk clusters interspersed with straighter, longer steps that, overall, were of similar complex form to Lévy-like movements in controls (**Figure 2B** and **5**). Overall, 26 paths were best fit by truncated power-laws (**Table 2**; **Supplementary file 3**; **Figure 5**). The exponential was the best fit for 13 tracks, while a further 15 were best fit by neither model (**Supplementary file 1**). No other model distribution tested provided better fits (**Supplementary file 4** and **6**).

Analysis showed that the 26 BL >$shi^{ts}$ larvae movement paths best fit by truncated power-law (Lévy) distributions had a mean exponent of 1.66 (± 0.50 S.D.; (**Figures 5** and **6**; **Table 2**), with CPD estimates for these exponent parameters predicting that 93% of the population would fall within the Lévy range, and 52% between $\mu$ exponent values of 1.5 and 2.5 (**Table 3**). The mean exponent was higher than BL/+ and $shi^{ts}$/+ larvae power-law exponents at both 22℃ and 33℃ (**Table 2**), but not significantly (**Figure 6**). Furthermore, computation of MSD confirmed similar long-range correlations (**Table 4**). Collectively, these results indicate that the movement path patterns of BL >$shi^{ts}$ larvae with blocked brain and SOG were also Lévy-like and statistically indistinguishable from those of BL/+

**Table 3.** Cumulative probability distributions.

Cumulative probability distributions of truncated power law exponents calculated from individual larva move-step length distributions across experimental treatments. T (℃) denotes different environmental temperature of treatment. Note that $\mu$ exponents of $shi^{ts}$/+ at 33 ℃ were not normally distributed and the treatment was not included in table (see Materials and methods).

| T (℃) | Treatment | Cumulative Probability Distribution (%) | | | |
|---|---|---|---|---|---|
| | | $\mu$ exponent range | | | |
| | | 1–3 | 1.25–2.75 | 1.5–2.5 | 1.75–2.25 |
| 22 | BL/+ | 93.2 | 71.2 | 41.3 | 17.1 |
| 33 | BL/+ | 90.4 | 59.6 | 26.4 | 8.1 |
| 22 | $shi^{ts}$ /+ | 98.4 | 76.8 | 33.6 | 8.0 |
| 33 | MB247/+ | 96.2 | 83.9 | 60.2 | 30.6 |
| 33 | BL > $shi^{ts}$ | 92.5 | 76.0 | 51.6 | 25.3 |
| 33 | BLsens > $shi^{ts}$ | 98.7 | 95.1 | 80.6 | 47.3 |
| 33 | MB247 > $shi^{ts}$ | 96.6 | 79.9 | 50.6 | 22.4 |
| 22 | BLsens > rpr, hid_control | 94.8 | 75.0 | 45.2 | 19.3 |
| 32 | BLsens > rpr, hid | 92.9 | 85.1 | 67.9 | 38.4 |

**Table 4.** MSD analysis results.

Mean and standard deviations for the values of alpha and R² from the mean squared displacement analysis. T (°C) denotes different environmental temperature of treatment.

| T (°C) | Treatment | No. of tracks | Mean α | SD | Mean $R^2$ | SD |
|---|---|---|---|---|---|---|
| 22 | BL/+ | 30 | 1.34 | 0.51 | 0.88 | 0.10 |
| 33 | BL/+ | 33 | 1.41 | 0.53 | 0.90 | 0.10 |
| 22 | $shi^{ts}$ /+ | 34 | 1.26 | 0.45 | 0.88 | 0.16 |
| 33 | $shi^{ts}$ /+ | 39 | 1.37 | 0.73 | 0.83 | 0.21 |
| 33 | MB247/+ | 23 | 1.40 | 0.81 | 0.84 | 0.12 |
| 33 | BL> $shi^{ts}$ | 26 | 1.21 | 0.55 | 0.81 | 0.24 |
| 33 | BLsens> $shi^{ts}$ | 11 | 0.94 | 0.78 | 0.67 | 0.25 |
| 33 | MB247> $shi^{ts}$ | 23 | 1.03 | 0.42 | 0.80 | 0.18 |
| 22 | New BLsens> $rpr,hid$_control | 62 | 1.17 | 0.49 | 0.81 | 0.20 |
| 32 | New BLsens> $rpr, hid$ | 23 | 1.30 | 0.53 | 0.80 | 0.14 |

and $shi^{ts}$/+ larvae without brain and SOG block at 33°C, and at an environmentally preferred temperature (22°C).

The path tortuosity analysis of BL/+, $shi^{ts}$/+ and BL >$shi^{ts}$ larvae showed that there was complex spatial clustering of intensive and extensive movements (*Figure 5*) and that in both treatments the frequency distributions of straightness-index cluster sizes across scales were best fitted by truncated power-law models (*Table 1*). Previous studies (*Sims et al., 2014*; *de Jager et al., 2012*) have identified that spatially nested composite Brownian (CB) walks may have parameters finely tuned to Lévy walks (*Reynolds, 2015*). To investigate this further, we compared larvae move-step distributions with three composite Brownian (CB) walk distributions comprising proportions of two, three, and four exponential distributions (CB2, CB3 and CB4; Materials and methods). Comparing truncated power-law, power, exponential and the three CB walks within the model set demonstrates that 13% of BL/+ larvae move-step distributions (8 of 63 distributions) were best fit by second or third-order CB walks, with two tracks (3.2%) best fit by pure power laws. In contrast, in $shi^{ts}$/+ larvae 92% of move step distributions (36 of 39 distributions) were best fit by a pure or truncated power law, with three best fit by second or third or CB walks. Similarly, 85% of BL >$shi^{ts}$ distributions (22 of 26 distributions) were best fit by truncated power laws over exponential and second, third or fourth order CB walks, with 4 (15%) best fit by a CB2 distribution (*Supplementary file 5*). This suggests that BL/+ movements were typified by Lévy-like patterns resembling multi-scale and hierarchically nested Brownian walk clusters, confirming the results found from the tortuosity analysis (*Table 1*), but less extensive in terms of a repeating pattern across increasing scales shown by the $shi^{ts}$/+ and BL >$shi^{ts}$ larvae. The strong support within the model set for truncated power laws in BL >$shi^{ts}$ larvae combined with a higher power law exponent may indicate a fundamental difference whereby the lack of brain and SOG inputs may alter movements in such a way as to more closely approximate optimal Lévy-like walks.

The mushroom body is the main associative centre in insects for processing sensory information of several modalities ultimately contributing to the selection of appropriate behavioural outputs (*Erber et al., 1987*; *Schürmann, 1987*). It has been shown that the cholinergic driver *Cha3.3*-Gal80, present in the BL line, blocks Gal4 activity in the anterior dorsolateral cells (eight neurons) of the mushroom body (*Tastekin et al., 2015*). Therefore, to eliminate the possibility of a potential contribution of neurons of the mushroom body, we used the specific MB247 driver line and impaired mushroom body activity using UAS-$shi^{ts}$ (*Thum et al., 2006*). In controls, we found 74% of MB247/+ larvae tracks, with intact mushroom bodies in the brain, were best fit by truncated power laws with a mean exponent of 1.74 (± 0.48 S.D.) over 1.33 orders of magnitude of the data (*Figure 6*; *Table 2*; *Supplementary file 3*, *4* and *6*), and a CPD predicting 96% within the Levy range and 60% between exponent values of 1.5 and 2.5 (*Table 3*). A truncated power law best fit to straightness index cluster sizes (*Table 1*) and an MSD α >1 also supported power law behaviour in the movement trajectories

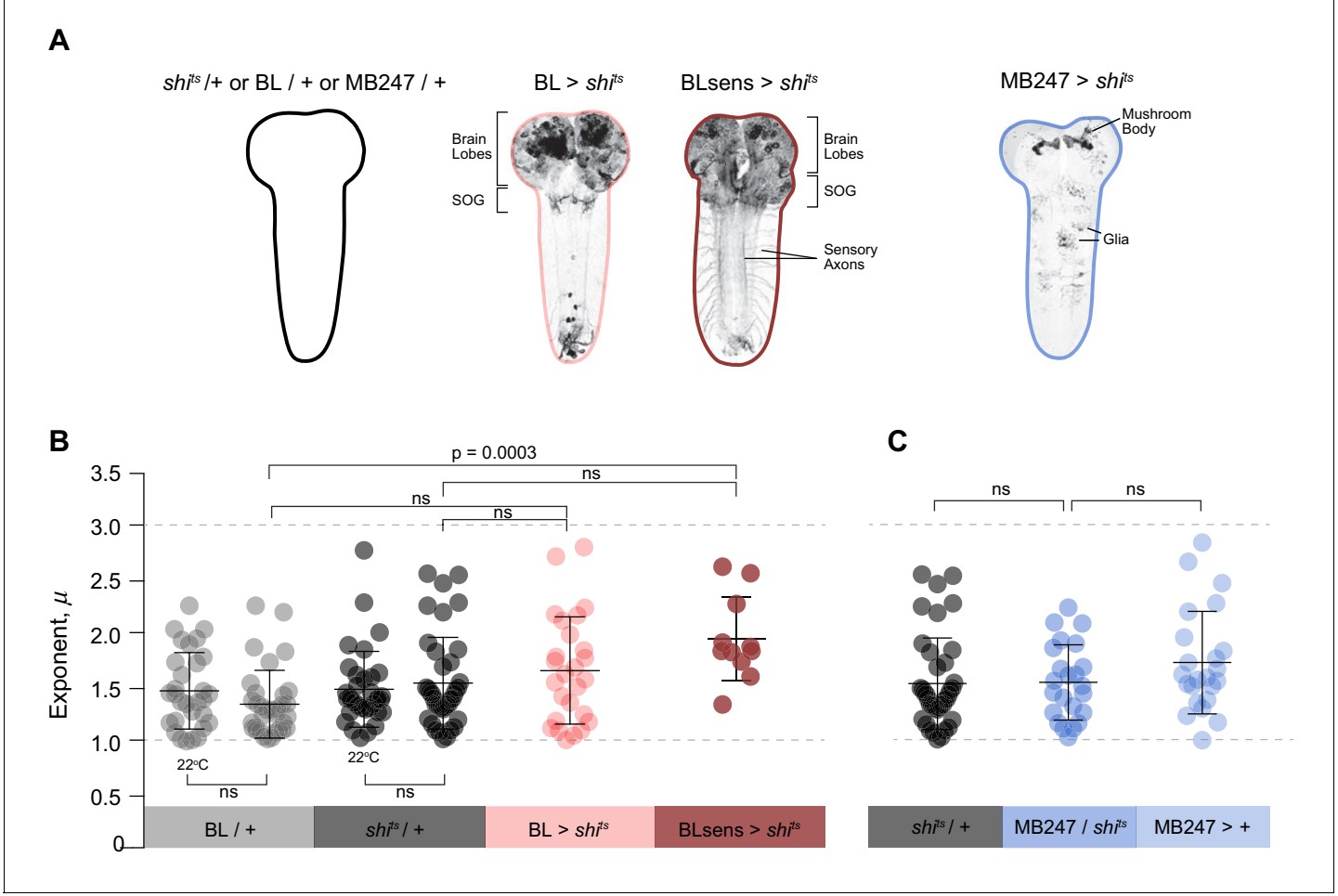

**Figure 6.** Summary data of truncated power law exponents across treatments. (A) Images and diagrams of the pattern of expression targeted in each genotype. (B) Comparison of $\mu$ exponents of the truncated Lévy power-law best model fits to exploration patterns of BL/ + and $shi^{ts}$/+ control larvae with intact nervous systems, BL >$shi^{ts}$ with blocked brain and a section of the SOG, and BLsens > $shi^{ts}$ with blocked brain, SOG and somatosensation, within the Lévy range (1 < $\mu$ ≤3; range bounds denoted by grey dotted lines). Scatter plot and mean (± SD) exponents: note the theoretically near-optimum search pattern ($\mu_{opt}$opt2) of BLsens > $shi^{ts}$ larvae demonstrated by the mean exponent ($\mu$) of 1.96. (C) Scatter plot and mean (± SD) of $\mu$ exponents of the truncated Levy power-law best model fits for the mushroom body experiment at 33˚C. The data distribution was not normal using D'Agostino-Pearson normality test. A Kruskal-Wallis test showed significant differences between the genotypes p=0.0002, $H_{(8)}$=28.27. Post doc Dunn's multiple comparisons relevant test results are shown on the graph.

(*Table 4*). For larvae with blocked mushroom bodies (MB247 >$shi^{ts}$), we found very similar results: 74% of track move-step length distributions were best fit to truncated power laws with a mean exponent of 1.56 (S.D. 0.35) over 1.39 orders of magnitude of data (*Figure 5*; *Table 2*; *Supplementary file 1*, *3*, *4* and *6*) and power-law distributed cluster sizes and long-range correlations in move step length time series (*Tables 1* and *4*). The mean exponent of MB247 >$shi^{ts}$ was not significantly different from its hemizygote control $shi^{ts}$/+, therefore the results indicate that active or inactive mushroom bodies in free-moving larvae had no major effect on the Lévy-like patterns of movement observed.

In conclusion, when brain processing of sensory information is impaired, extended substrate exploration emerges as multi-scale movement paths resembling truncated Lévy walks, strongly supporting the intrinsic hypothesis.

## Optimal searching is generated autonomously by thoracic and abdominal circuits

To further test the intrinsic hypothesis, we evaluated whether the Lévy-like move patterns were generated by the thoracic and abdominal CPG by tracking BLsens > $shi^{ts}$ (elav-Gal4, tsh-Gal80/+,+; UAS-$shi^{ts}$/+) larvae where synaptic transmission in the brain, SOG, and all sensory neurons is affected at the restrictive temperature (proportion of activity left at 33°C compared to activity at 23°C was −0.01 ± 0.22 S.D.; *Figure 4*) (*Berni et al., 2012*; *Berni, 2015*). We tracked 30 BLsens > $shi^{ts}$ larvae and, as expected (*Berni et al., 2012*; *Berni, 2015*), we found that these larvae moved more slowly than the other treatments. It was evident that tracks had fewer long move steps than observed for the other larvae (lower straightness index values; *Figure 5*), with trajectories instead being characterised by having a high frequency of highly spatially intensive movements occurring intermittently along the path (*Figure 5*). In total, 21 tracks had sufficient computed move step lengths to support robust analysis (*Supplementary file 1*).

Analysis showed 11 (of 21 tracks) had best fits to truncated power-law distributions (*Table 2*; *Supplementary file 3*; *Figure 5*), with two tracks best fit by the exponential distribution and eight tracks best fit by neither distribution (*Supplementary file 1*). Despite the tracks showing exponential fits having sufficient computed move step lengths, these larvae did not undertake extensive movements instead spending major portions of the 1 hr trial within a small area. The 11 truncated power law fits had a mean exponent of 1.96 (± 0.39 S.D.) (*Table 2*), higher than the mean exponent of the BL >$shi^{ts}$ larvae (*Figure 6*), and closer to the theoretical optimum, $\mu_{opt} \cong 2$ (*Viswanathan et al., 1999*; *Viswanathan et al., 2011*; *Humphries and Sims, 2014*; *Wosniack et al., 2015*) and over a broad range of move step lengths (mean orders of magnitude, 1.20 ± 0.23 S.D., $n$ = 11) (*Table 2*; *Figure 5*). The CPD predicts 81% of randomly chosen BLsens > $shi^{ts}$ exponent values would fall between 1.5 and 2.5, and 47% between 1.75 and 2.25 (*Table 3*), indicating the majority of exponents were close to $\mu_{opt}$ = 2. No other model distributions provided better fits to the data (*Supplementary file 4* and *6*). Long-range correlations in MSD of BLsens > $shi^{ts}$ larvae trajectories were present with $\alpha \sim 1$ (*Table 4*). As with the BL/+ and BL >$shi^{ts}$ larvae, we also found the frequency distributions of straightness index cluster sizes were best fit by truncated power laws over a mean of 1.4 orders of magnitude of the data (*Table 1*). Furthermore, 55% of tracks were best fit to truncated or pure power laws rather than exponential or composite Brownian walks (*Supplementary file 5*), similar to the results found for BL >$shi^{ts}$ larvae.

To validate our results we used a second approach; instead of blocking synaptic transmission with UAS-$shi^{ts}$ we induced specific neuronal death early during larval development in the brain lobe, SOG and sensory system (*Figure 7A–D*). We generated a new and stronger BLsens driver line (BLsens$^{R57C10}$) by replacing elav-gal4 with R57C10-gal4 and crossed it to the proapoptotic genes activators (UAS-rpr and UAS-hid) using tubulin-gal80$^{ts}$ to restrict the moment of expression (UAS-rpr, UAS-hid/+,+; elav-Gal4, tsh-Gal80/+, tub-Gal80$^{ts}$/+; Materials and methods; *Figure 7A–D*). After determining a set of temperatures that kill the neurons and allow the larvae to reach the third instar, we performed behavioural experiments in the same way as the other lines tested. At the control temperature of 22°C with no induced apoptosis, we found 80% of BLsens > rpr, hid larvae had move step length distributions best fit by truncated power laws with an exponent of 1.51 over 1.71 orders of magnitude of the data (*Figure 8*; *Table 2*; *Table 3*; *Supplementary file 3*), and cluster and MSD analyses confirming power law behaviour (*Tables 1* and *4*). Of the 77 tracks analysed, five were best fit by the exponential and 10 were best fit by neither distribution. For BLsens > rpr, hid larvae tracked at the restrictive temperature of 32°C, inducing apoptosis of brain plus sensory neurons, the trajectories were generally more spatially restricted for the majority of individuals. Although this resulted in more exponential fits to step-length frequency distributions (10 tracks from 62 retained), it was striking that 37% of distributions were best fit by truncated power laws (*Table 2*; *Supplementary file 1*; *Figure 7* and *8*). Moreover, the mean exponent for BLsens > rpr, hid was 2.14, with CPD analysis demonstrating 68% of population exponents fall within the $\mu$ value range of 1.5 to 2.5, and 38% between 1.75 and 2.25 (*Table 3*). All other analyses supported the presence of Lévy-like walks in movement patterns best fit by truncated power laws (*Tables 1* and *4*). Interestingly, the power law exponent of BLsens > rpr, hid larvae movement patterns at 32°C, killing most brain and sensory neurons, was significantly higher and closer to the theoretical optimum than controls at 22°C (*Figure 7B, D, E*). This was similar to the result found for power law exponents of BLsens > $shi^{ts}$

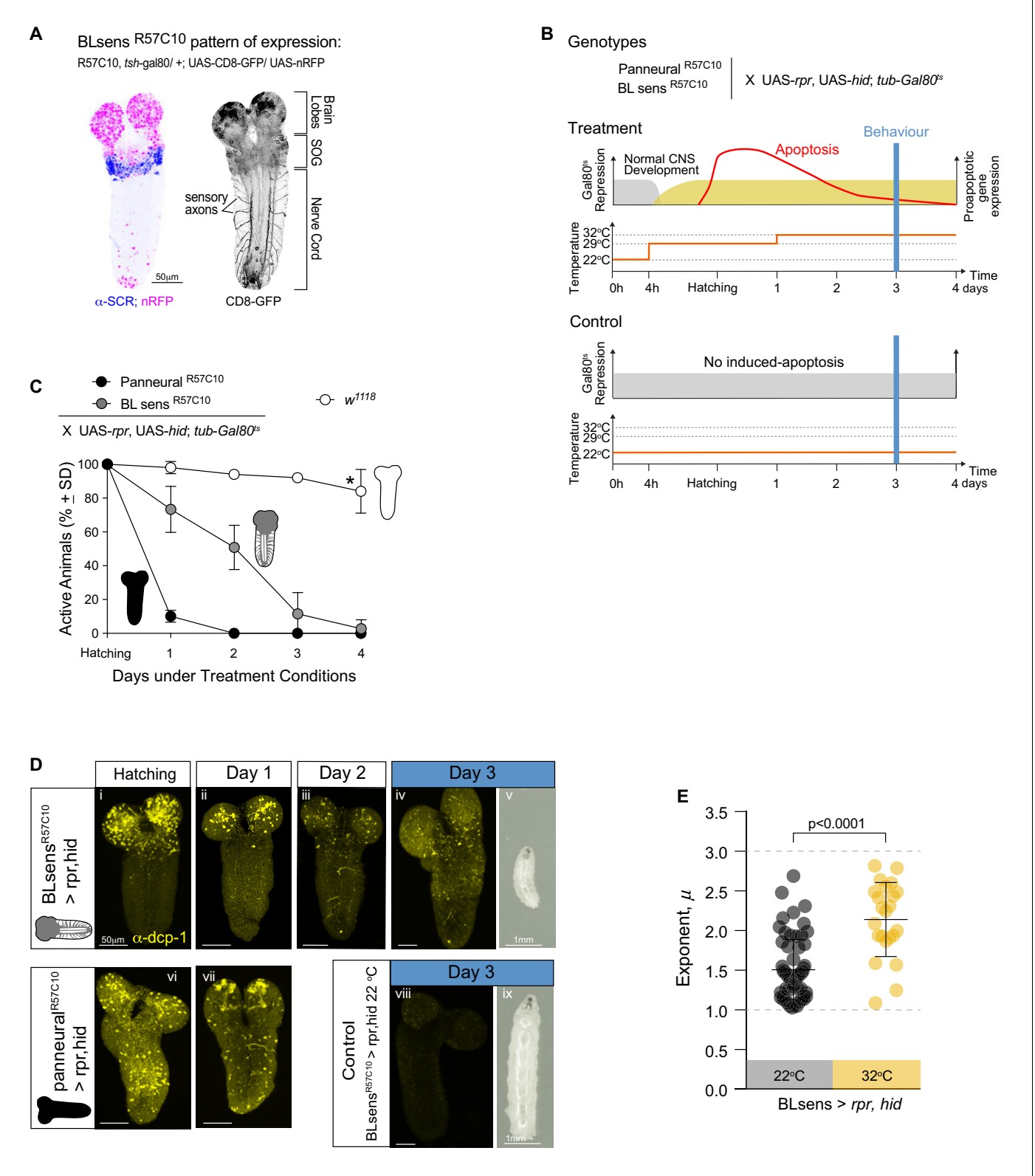

**Figure 7.** Near optimal Lévy search in larvae after apoptosis in the brain lobes. (**A**) Expression pattern of the BLsens[R57C10] line. Gal4 is highly expressed in the brain lobes and SOG until the anterior boundary of SCR expression. (**B**) Schematic of the experimental design. Each one of the genotypes was subjected to the treatment protocol, which induces the expression of proapoptotic genes (*rpr* and *hid*) in mid embryonic stages resulting in apoptosis when the larvae hatch. Higher levels of apoptosis were induced by increasing temperature to 33°C 24 hr after hatching. Control animals were kept

*Figure 7 continued on next page*

*Figure 7 continued*

under non-activating (22°C) conditions. The behavioural experiments were conducted at day 3. (**C**) Quantification of active larvae in the treatment conditions. $w^{1118}$ was used as a control. Asterisk denotes that at day 4 a few larvae have moulted. (**D**) Apoptosis and larval size. The expression of Death Caspase-1 (anti-CPD-1 in yellow) was used as a readout of cell death. Nervous systems were evaluated at hatching (**i, vi**), at day 1 (**ii, vii**) and at day 2 (**iii**) and day 3 (**iv,viii**). Neuronal death is observed in the brain lobes of BLsens[R57C10] > *rpr,hid* larvae and in the entire nervous system of R57C10 > *rpr,hid* animals at hatching and day 1. At day 2, R57C10 > *rpr,hid* were all dead indicating that a high level of apoptosis was reached. The number of neurons ongoing apoptosis started declining in BLsens[R57C10] > *rpr,hid* larvae brains until day three when CPD-1 was only observed in granules. Larvae with brain lobes death (**v**) are smaller than control animals (**viii**) likely due to decrease food intake. (**E**) Scatter plot and average exponents (± SD) of the truncated Levy power-law best model fits to exploratory pattern. R57C10 > *rpr,hid* have a mean exponent (μ) of 2.1 close to the theoretically near-optimum search pattern ($\mu_{opt}$opt2). The data distribution was not normal using D'Agostino-Pearson normality test. A Mann Whitney test showed significant differences between the BLsens[R57C10] > *rpr,hid* control animals raised at 22°C and the treatment with induced apoptosis. $U = 222.5$, $n_{control} = 62$, $n_{treatment} = 23$, $p < 0.0001$.

compared to *shi[ts]*/+ control larvae (*Figure 6*). Therefore, the results obtained with proapoptotic driver lines were similar to those seen in the experiments with UAS-*shi[ts]*, which reinforces our finding that even without brain function larvae explore their environment with movement patterns well approximated by Lévy walks.

In summary, it is striking that larvae with dramatically disrupted brain, SOG or sensory neural activity were still able to undertake complex, multi-scale movement paths that approximate a Lévy-like search pattern, a theoretically optimal strategy for 'blind' foragers searching for sparse target resources in unknown locations. Our result is the strongest evidence yet for an intrinsic, autonomous mechanism underlying Lévy-like patterns of movement in an animal.

## Discussion

Intrinsic Lévy walks in animal searches have been hypothesised based on experiments showing Lévy-like movements of insects (*Maye et al., 2007*; *Reynolds, 2015*; *Cole, 1995*; *Bazazi et al., 2012*) and molluscs (*Kölzsch et al., 2015*; *de Jager et al., 2014*) exposed to minimal external stimuli under controlled conditions. Similarly, we have demonstrated that freely moving *Drosophila* BL/+, *shi[ts]*/+ and MB247/+ control larvae tracked over large relative time and space scales with minimal sensory

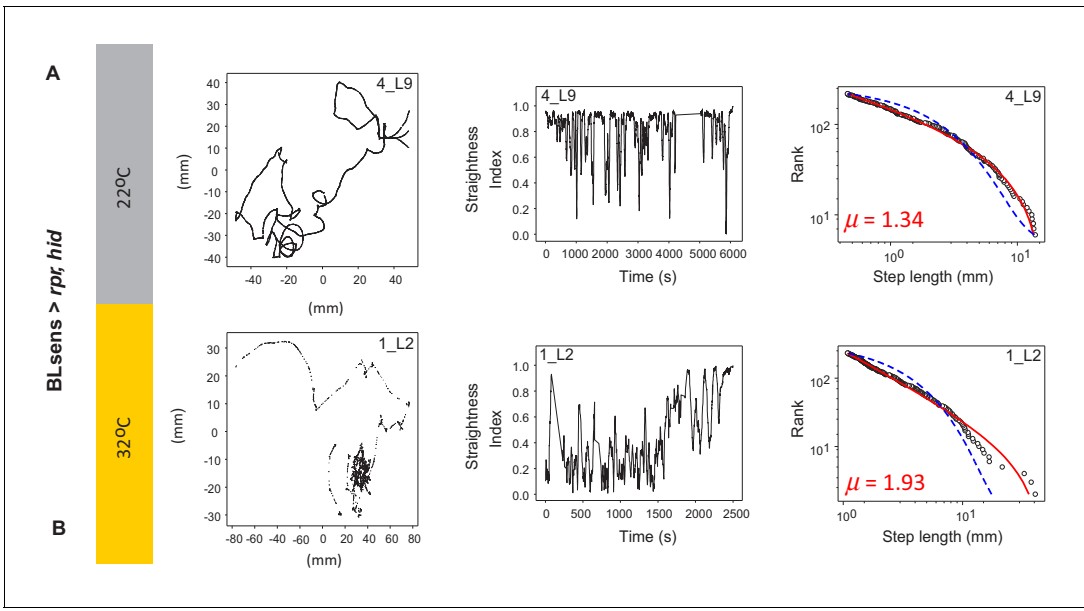

**Figure 8.** Example tracks in larva after apoptosis in the brain lobes. (**A**) Control BLsens > *rpr,hid* larvae at 22°C show movement patterns, path complexity and truncated power law best fits that are similar to (**B**) BLsens > *rpr,hid* larvae at 32°C with apoptosis in the brain lobes. Note that the larva path in (**B**) has a μ exponent of 1.93 that lies closer to the theoretical optimum than the control, indicating an intrinsic neural origin of Lévy-like search patterns in *Drosophila* larvae. See *Table 2* for mean exponents.

inputs (visual, olfactory, gustatory) and homogeneous temperatures at 22°C and 33°C display exploration (search) patterns well approximated by Lévy-like searches. All previous observations, even in favour of the extrinsic hypothesis (*Censi et al., 2013*) were demonstrated in animals with intact nervous systems, thus the influence of the sensory inputs from the environment could not unambiguously be asserted.

Our finding that BL >*shi^{ts}* larvae with strongly impaired brain processing and part of SOG function explore with multi-scale, Lévy-like movement patterns provides the first strong evidence that a Lévy-like pattern was not an emergent property of interactions with fractally distributed resources or sensory landscapes (rejecting the extrinsic hypothesis). In contrast, we show that substrate exploration resembling a Lévy walk pattern is an intrinsic routine for exploration (support for the intrinsic hypothesis). That BLsens >*shi^{ts}* larvae exhibit multi-scale, Lévy-like patterns and heavy-tailed distributions – via forward crawls interspersed by pauses then turns – indicates that multi-scale movement patterns are generated spontaneously by the central networks of the thoracic and abdominal system that operate independently of input from either the brain or the sensory system. This was confirmed in larvae BLsens >*rpr, hid* larvae expressing proapoptotic gene activators that kill brain and sensory neurons, which were also free-moving with patterns from long trajectories well approximated by Lévy-like walks. Therefore, we conclude that multi-scale, Lévy-like search patterns arise intrinsically and autonomously in *Drosophila* larvae.

We also found evidence for intrinsic Lévy-like patterns that are theoretically optimal for sparse resource target encounter in patchy landscapes. Model fits to movement patterns gave power law exponent values of $\mu = 1.66$ for BL >*shi^{ts}* and $\mu = 1.96$ for BLsens > *shi^{ts}* larvae, values close to the theoretically optimal Lévy search pattern of $\mu \sim 2$. Recent simulations demonstrate that Lévy walks with $\mu \sim 2$ are optimal for a much broader range of resource densities and distributions than generally realised (*Humphries and Sims, 2014*; *Wosniack et al., 2015*). In addition, numerous studies show that Lévy walks with power law exponents $< 2$ yield more ballistic searches which are optimal for finding target resources more distantly located (*Viswanathan et al., 2011*; *Humphries and Sims, 2014*). It is also apparent that Lévy searches with $\mu \sim 1.5$, and similar to exponents shown by BL/+ and *shi^{ts}*/+ larvae (1.35 and 1.54 respectively) at the restrictive temperature of 33°C, have in simulations been shown to have the lowest variance in foraging efficiency compared to theoretically optimal Lévy foragers, in addition to having low 'famine' durations that are only slightly greater than for $\mu \sim 2$ Lévy searches (*Humphries and Sims, 2014*). Taken together, this suggests *Drosophila* larval exploration patterns were similar to theoretically optimal Lévy searches for efficient location of sparse and distant resources.

The restrictive temperature of 33°C did not affect patterns of movements in larvae because there was no significant difference in power law exponents for BL/+ or *shi^{ts}*/+ at 33°C or 22°C, a preferred temperature for these larvae. The difference between exponents of BL/+ and *shi^{ts}*/+ larvae with BL >*shi^{ts}* larvae may be as a result of sensory responses of BL/+ and *shi^{ts}*/+ search patterns to minimal external cues likely to have been present in our experimental set-up. For these larvae, the lack of food cues may have resulted in a higher proportion of longer steps (more extensive searching) giving rise to the lower exponent. This implies that the theoretically near-optimal search pattern of BL >*shi^{ts}* larvae emerges from central pattern generation in the absence of external sensory inputs.

Precisely how intrinsic, autonomous Lévy-like search patterns are generated in the nervous system of *Drosophila* larvae was not investigated in this study. Recent studies however, have localised the networks controlling the elements of the behaviour sequence (*Berni et al., 2012*; *Berni, 2015*) which provides some insight. Exploratory behaviour is already present in newly hatched *Drosophila* larvae consisting of straight-line forward crawls driven by peristaltic contractions that are interrupted by pauses, turns and redirected forward crawls (*Berni et al., 2012*). Genetic dissection of neural circuitry of this exploratory behaviour shows that larval straight-line crawls rely on symmetrical output of the thoracic and abdominal network, whereas thoracic segments generate the asymmetric activity resulting in a turn (*Berni, 2015*). An intermittent interruption to crawling determined the time-dependent alternation between crawls and turns, but it was not known which part of the system was responsible or whether it was dependent on a fluctuating property of the whole network such as excitation level (*Berni, 2015*).

A Lévy movement pattern will emerge when straight-line, ballistic movement is interrupted by pauses with intervals timed stochastically according to a heavy-tailed (e.g. power-law) distribution of pause times. It is possible that Lévy movement we observed in *Drosophila* larvae could arise from

the temporal structure of pauses arising from a 'Lévy timing mechanism' (LTM) within the thoracic and abdominal network. We propose that the LTM is generated by temporally scale-free changes in neuronal activity of the circuit controlling the transition between straight-line crawls and turns. Scale-free temporal structuring of activity has been observed from single neurons to circuits and is a wide-spread feature of nervous systems (*Mazzoni et al., 2007*), suggesting that these patterns of activity could underlie the generation of a Lévy movement. Furthermore, an LTM arising within the thoracic and abdominal neural network of free-moving *Drosophila* larvae could be analogous to hypothetical 'fractal reorientation clocks' in intermittent locomotion (*Bartumeus and Levin, 2008*). It was proposed in a simulation study that a scale-free (Lévy) distributed pattern of times between pauses (stops or strong changes in speed) punctuating the movement path of a model organism results in Lévy intermittence that may come from reorientation mechanisms capable of organising directional persistence on time (*Bartumeus and Levin, 2008*). The neural coding processes underpinning a potential mechanism such as this remains to be elaborated precisely, but a link between fractal timing dynamics and propagation patterns in neural networks has been proposed from experimental observations (*Beggs and Plenz, 2003*). Patterns of spontaneous activity in cortical networks can occur as cascades or 'avalanches', where it is proposed that the activity of an individual neuron causes activities in other units in turn, thereby initiating a neuronal avalanche through the network (*Beggs and Plenz, 2003*). The spatial and temporal distributions of neuronal avalanches appear well described by power laws (*Beggs and Plenz, 2003*), which indicates the network to be in a critical state such that its system dynamics occur at many different spatio-temporal scales (scale-free) (*Beggs and Plenz, 2003*). Furthermore, a neural coding scheme based on bursty, multifractal time-series patterns of spike potentials recorded from rat hippocampus during foraging has been proposed to generate neurologically based superdiffusive (Lévy) search patterns (*Gutiérrez and Cabrera, 2015*). In this context, our empirical study provides some support for the idea of an LTM or 'fractal reorientation clock' organising movement paths. Indeed, scale-free or fractal initiators of signalling pathways underlying movement patterns in the absence of prior information about resource location may be common in nature across diverse taxa, from individual cells to animals (*Maye et al., 2007*; *Korobkova et al., 2004*; *Proekt et al., 2012*; *Namboodiri et al., 2016*).

Our results indicate that multi-scale movement paths that appear complex are generated intrinsically and we show that a relatively simple neural network generated within the thoracic and abdominal nervous system is the principal driver. That free-moving *Drosophila* larvae without brain and sensory function crawl and explore their environment with apparently complex search patterns that approximate optimal Lévy walks should perhaps not be considered so surprising since it is well established that higher animals can have spared function following decerebration. In vertebrates such as rats for example, much behavioural capacity is spared following decerebration at the mesencephalic and precollicular levels with preservation of a number of complex and well-coordinated behaviour patterns (*Woods, 1964*; *Lovick, 1972*). These include locomotion and components of feeding, grooming and defensive behaviour (*Woods, 1964*; *Lovick, 1972*). These results for vertebrates clearly suggest that the neural circuitry for many elements of species-typical behaviours can be organised in lower regions of the neuraxis and that these can be integrated to form complex patterns of behaviour. Thus, the intrinsic Lévy-like search patterns we identify in *Drosophila* with no brain or sensory functions can be viewed as an underlying 'free running' optimal search pattern that is modified by environmental stimuli. We propose that when sensory inputs are received to the brain, such as olfactory stimuli, descending signals (*Tastekin et al., 2015*) to the thoracic and/or abdominal circuits modulate neural activity from Lévy-like patterns to deterministic patterns of movement driven by environmental information, e.g. chemotaxis (*Gomez-Marin et al., 2011*).

Multi-scale movement patterns exhibited by brain and sensory blocked larvae are the intrinsic output of the thoracic and abdominal nervous system and appear finely tuned to theoretically optimal Lévy walks, which suggests selection pressure for Lévy walk characteristics (*Reynolds, 2015*; *Sims et al., 2014*). Indeed, comprehensive computer simulations have confirmed that Lévy walks arise as the optimal search strategy in a wide range of different environments, providing the density of resources is sparse and the searcher's information about resource location is limited to its close vicinity (*Wosniack et al., 2017*). This has been suggested to provide the strongest theoretical evidence underpinning the evolutionary origins of empirically observed Lévy walks (*Wosniack et al., 2017*). Thus, because searching behaviour is crucial to mobile animals' success (e.g. for food and mate location), the simple neural circuitry giving rise to complex, intrinsic Lévy-like searches that are

theoretically optimal in diverse patchy landscapes (*Humphries and Sims, 2014*; *Wosniack et al., 2015*; *Wosniack et al., 2017*) may have adapted early in the evolution of nervous systems. Central-ised nervous systems comprised of cerebral ganglia were likely present in the Ediacaran (*North-cutt, 2012*) and brains and trunk ganglia are evident in early Cambrian arthropods (*Tanaka et al., 2013*). Early presence of Lévy-like behavioural outputs from simple neural circuitry, such as that observed for *Drosophila* larvae, could help explain the widespread occurrence of multi-scale, Lévy-like behaviour among widely separated modern taxa, such as fish and birds (*Sims et al., 2008*; *Humphries et al., 2012*; *Humphries et al., 2010*), that have a last common ancestor over 450 mil-lion years ago. We cannot exclude, however, the possibility for convergent evolution of similar movement patterns across diverse taxa. Nevertheless, an advantage as beneficial for resource acqui-sition by 'blind foragers' as Lévy search patterns (*Viswanathan et al., 2011*; *Humphries and Sims, 2014*; *Wosniack et al., 2015*), together with our finding of intrinsic neuronal generation of multi-scale, Lévy-like behaviour for substrate exploration in an animal, presents the intriguing possibility that search strategies evolved to exploit Lévy patterns.

# Materials and methods

## Key resources table

| Reagent type (species) or resource | Designation | Source or reference | Identifiers | Additional information |
|---|---|---|---|---|
| Strain, strain background (*Drosophila melanogaster*) | w; *elav*-GAL4, *tsh*-GAL80/ +,+; *cha3.3*-GAL80, UAS-EGFP/ + | DOI: 10.1016/j.cub.2012.07.048 | NA | BL/+ |
| Strain, strain background (*Drosophila melanogaster*) | w; *elav*-GAL4, *tsh*-GAL80/ +,+; *cha3.3*-GAL80, UAS-EGFP / + | DOI: 10.1016/j.cub.2012.07.048 | NA | BL > + |
| Strain, strain background (*Drosophila melanogaster*) | w; *elav*-GAL4, *tsh*-GAL80/ +,+; UAS-EGFP / + | DOI: 10.1016/j.cub.2012.07.048 | NA | BLsens > + |
| Strain, strain background (*Drosophila melanogaster*) | 20XUAS-IVS-Syn21-Shibire-ts1-GFP-p10 | Gift from G. Rubin | NA | |
| Strain, strain background (*Drosophila melanogaster*) | MB247-GAL4 | Gift form J. Ng | NA | |
| Strain, strain background (*Drosophila melanogaster*) | UAS-*rpr*, UAS-*hid* | Gift from from M. Landgraf | NA | |
| Strain, strain background (*Drosophila melanogaster*) | UAS-myrRFP | Gift from from M. Landgraf | NA | |
| Strain, strain background (*Drosophila melanogaster*) | P{GMR57C10-GAL4}attP2 | Bloomington Drosophila Stock Center | RRID:BDSC_39171 | In attp40 |
| Strain, strain background (*Drosophila melanogaster*) | P{tubP-GAL80[ts]}2 | Bloomington Drosophila Stock Center | RRID:BDSC_7017 | |
| Chemical | Poly-L-Lysine hydrobromide | Sigma | P2524 | |

*Continued on next page*

*Continued*

| Reagent type (species) or resource | Designation | Source or reference | Identifiers | Additional information |
|---|---|---|---|---|
| Antibody | Mouse monoclonal anti-SCR | Development Studies Hybridoma Bank (DSHB), IA, USA | RRID:AB_528462 | 1: 20 dilution |
| Antibody | Chicken polyclonal anti-GFP | abcam | ab13970 | 1/2000 dilution |
| Antibody | Rabbit polyclonal anti-Cleaved Drosophila Dcp-1(Asp216) | Cell Signaling Technology | #9578 | 1/100 |
| Antibody | Donkey polyclonal Alexa568 anti-mouse | Invitrogen | A10042 | 1/500 |
| Antibody | Donkey polyclonal CF633 anti-rabbit | Biotum | BT20125 | 1/500 dilution |
| Antibody | Goat polyclonal Alexa488 anti-chicken | Biotum | BT20020 | 1/500 |
| Software | FIMTrack | https://www.uni-muenster.de/PRIA/en/FIM/download.shtml | NA | FIMTrack_v2_X64_MacOS |
| Software | MBA MLE Analysis | http://dx.doi.org/10.1111/2041-210X.12096 | NA | |
| Software | clampfit | https://moleculardevices.app.box.com/s/l8h8odzbdikalbje1iwj85x88004f588 | NA | |

## Animals

BL/ +: Control BL (for brain lobes) animals that allow manipulation of the brain lobes and a section of the supraoesophageal ganglion (SOG).

BL >*shi*$^{ts}$: The larvae express in the brain lobes and a section of the SOG a dominant negative temperature-sensitive form of Shibire, UAS-*shi*$^{ts}$, that blocks synaptic transmission at the restrictive temperature (***Thum et al., 2006***; ***Kitamoto, 2001***). The new stronger pJFRC101 version 20XUAS-IVS-Syn21-Shibire-ts1-GFP-p10 in attP2 (***Pfeiffer et al., 2012***), was used since it blocks synaptic transmission at lower temperatures (above 30˚C; ***Figures 2*** and ***3***) than the one previously used (above 34˚C) (***Berni et al., 2012***). In our experiments, at 33˚C, any possibility of environmental sensory integration by the brain is removed (***Berni et al., 2012***). These larvae were tracked to test whether Lévy-like movement patterns were intrinsic.

BLsens > *shi*$^{ts}$: Larvae where synaptic transmission in the brain, SOG, and all sensory neurons is blocked at the restrictive temperature (***Berni et al., 2012***), were tracked to test whether Lévy-like move patterns were generated by the CPGs located in the thoracic and abdominal central nervous system.

## Treatment Genotypes

BL/ +: w; *elav*-GAL4, *tsh*-GAL80/ +,+; *cha3.3*-GAL80, UAS-EGFP/ +

BL >*shi*$^{ts}$: w; *elav*-GAL4, *tsh*-GAL80/ +,+; *cha3.3*-GAL80, UAS-EGFP/20XUAS-IVS-Syn21-Shibire-ts1-GFP-p10

BLsens > *shi*$^{ts}$: w; *elav*-GAL4, *tsh*-GAL80/ +,+; UAS-EGFP/20XUAS-IVS-Syn21-Shibire-ts1-GFP-p10

BLsens$^{R57C10}$ > rpr, hid: y-, w-, UAS-hid, UAS-rpr/ +; R57C10-GAL4, *tsh*-GAL80, UAS-myrRFP/ +; *tub*-Gal80$^{ts}$/+

## Stock source

20XUAS-IVS-Syn21-Shibire-ts1-GFP-p10 in attP2 was a gift of G. Rubin; MB247-Gal4 of J. Ng, y-, w-, UAS-hid,UAS-rpr; UAS-myr-RFP of M. Landgraf; *elav*-GAL4 (BL8765), R57C10 (BL39171) and *tub*-Gal80$^{ts}$ (BL7017) were from Bloomington Stock Center.

## Experimental treatments

Eggs were collected overnight on normal food plates from 6 females and six males to avoid over-crowding. Larvae were grown on abundant yeast paste at 22°C, a temperature where $shi^{ts}$ is not activated, for 4 days (*Kitamoto, 2001*). First, we tracked BL/+ ($n = 30$) and $shi^{ts}$/+ larvae ($n = 30$) at 22°C to determine movement patterns at an environmentally relevant temperature optimum for *Drosophila* movement and feeding. Second, we tracked early 3rd instar (4 day old) larvae in six experimental groups at 33°C, the $shi^{ts}$ activation temperature: BL/+, $shi^{ts}$/+, control animals; MB247/+, control; BL >$shi^{ts}$, brain and SOG synaptic activity block; BLsens > $shi^{ts}$, no brain, or SOG or body wall exteroception or proprioception; MB247 >$shi^{ts}$, mushroom body activity block. Thirty individuals were tracked per experiment in three trials ($n = 10$ larvae per trial). A further 10 larvae ($shi^{ts}$/ + at 33°C) were tracked at higher temporal rate of video recording (15 Hz) (see *Larva Tracking*). Overall, 250 individual larvae were tracked across these treatments. During the experiment, the temperature was monitored with a thermocouple probe placed on top of the agar in one corner of the arena attached to a multimeter (Uni-Trend UT60-E).

## Proapoptotic experiments

A protocol was generated to kill as many neurons as possible in the brain lobes and sensory system while allowing the larvae to reach third instar (*Figure 7B*). Eggs were collected for 4 hr on normal food plates at 22°C then transferred at 29°C. The following day, newly hatched larvae from the correct genotype were selected and groups of 10 larvae were transferred to new plates with abundant yeast paste. 24 hr later they were transferred to 32°C until reaching third instar larvae. To confirm the efficiency of the manipulation the number of active, sick and almost paralysed, or dead larvae in each plate was counted every day. A 100% death in R57C10 > rpr, hid, tubgal80$^{ts}$ larvae guaranteed that the level of expression of RPR and HID is sufficient to kill neurons (*Figure 7C*; n = 30 to 40 larvae in four trials). Six control treatment trials ($n = 10$ larvae per trial) were undertaken and treated in the same way but kept at 22°C. Seven trials ($n = 10$ larvae per trial) were carried out at the activation temperature (32°C). Therefore, 130 larvae were tracked in proapoptotic experiments.

## Immunohistochemistry

Nervous systems of early third instar larvae were dissected in PB (100 mM NaH$_2$PO$_4$/Na$_2$HPO$_4$) pH 7.2, transferred to a polylysine-coated cover slip fixed with 4% formaldehyde in PB for 20 min at room temperature (RT) and rinsed in PBS plus 0.3% Triton X-100 (PBT) 3 × 15 min. Specimens were then incubated with anti-SCR 1/20 (Developmental Studies Hybridoma Bank, USA); chicken anti-GFP 1/2000 (Abcam) or rabbit anti-DCP- asp216 1/100 (Cell Signaling) in PBT overnight at 4°C in a wet chamber, washed in PBT 4 × 15 min, and incubated with secondary antibodies at 1/500 in PBT for 3 hr at RT: Alexa568 anti-Mouse (Invitrogen); CF633 anti-rabbit and Alexa488 anti-Chicken (Biotum). Secondary antibodies were washed 4 × 30 min in PBT and specimens were mounted in Vectashield (Vector Laboratories) between two aluminium-foil spacers, to avoid distortion of nerve cords, under number one cover glasses. Image stacks were captured on a Leica TCS-SP-5 confocal microscope.

## Electrophysiology recordings

Third instar larvae were dissected and pinned out in a sylgard-coated Petri dish filled with physiological saline (135 mM NaCl, 5 mM KCl, 4 mM MgCl2.6H2O, 2 mM CaCl2, 5 mM N Tris (hydroxymethyl) methyl-2- aminoethanesulfonic acid and 36 mM sucrose, pH 7.15 [*Berni, 2015*]). Suction electrodes were pulled from thin-walled borosilicate glass (1.00 × 0.75 mm 4', A-M Systems Inc; Harvard Apparatus). To obtain a pipette tip with approximately the same or slightly smaller diameter than the individual brain lobe the pipette tip was cut with scissors under a microscope. The suction electrode was gently approached until touching the brain lobe and suction was applied until an increase in baseline activity was observed (see *Figure 4C*). A reference electrode was placed in the dish saline. After a stable recording was obtained, reference temperature (RT) was recorded (BAT-7001H thermometer with PT-6 sensor, Physitemp) and hot buffer was immediately superfused until target temperature was attained (33°C ± 1.0). Activity was recorded for several minutes while the lobes slowly reached recovery temperature (<29°C). Extracellular activity was acquired with an extracellular amplifier (A-M System 1700), filtered (Low cut-off, 1 Hz; high cut-off, 1 kHz) and digitalized (Digidata 1320A, Axon Instruments). Individual traces were analyzed with Clampfit (plamp10, Axon Instruments). Activity

was measured for 10 s after the temperature check. To quantify the brain lobe response to temperature changes, first, baseline (before suction) was subtracted from all recordings. Next, brain activity at the target temperature (33°C ± 1.0 S.D.) was normalised to that at reference temperature (≅23°C). This renders an index with value of 1 when no change respect to RT is observed, and zero when all brain activity is abolished.

## Larva tracking

In the majority of studies examining the behaviour patterns of *Drosophila*, including those with BL larvae, trials for tracking movement are generally short (1–10 min; e.g. *Berni et al., 2012*; *Vogelstein et al., 2014*). However, when testing for the presence of multi-scale behaviour such as Lévy search patterns, longer tracks are required because the underlying move-step frequency distribution of an animal's normal movement routine must be sampled thoroughly to increase the likelihood of detecting rare longer move steps, if they are present (*Sims et al., 2008*; *Humphries et al., 2012*; *Humphries et al., 2010*; *Humphries et al., 2013*). Longer track times require larger areas as larvae will likely move further in the trial time. For robust statistical detection of the macroscopic properties of a search pattern it is important for an organism's path to be as unbounded as possible, that is, for collision with the edges of the arena to be minimised. Edge collision will truncate move step lengths (distance between consecutive turns) introducing potential bias against the chances of recording long move steps that are characteristic of some movement search patterns, e.g. Lévy walks (*Humphries et al., 2010*). Consequently, track sections recorded after an edge collision were analysed as separate tracks and visual inspection was used to remove tracks where the larva simply travelled along the arena edge. Therefore, in this study each trial lasted 1 hr in which we tracked 10 young third instar larvae within a large 240 × 240 mm arena relative to the total body length of tracked larvae (mean, 2.37 mm ± 0.57 S.D.; $n$ = 90). Each larva was video tracked once. In addition, we tested for significant differences between μ values (summarising movement patterns) before and after collisions with other larva and arena edge before reorientation in both BL/+ and *shi*[ts]/+ treatments at both 22°C and 33°C (results given in *Figure 2—figure supplement 3*).

Groups of 10 early third-instar larvae were washed to remove traces of food and allowed to dry crawl for 2 min on a clean agar-coated plate (5 cm diameter) before being transferred to the arena. The exploratory behaviour was monitored using a Frustrated Total Internal Reflection (FTIR)-based Imaging Method for high throughput locomotion analysis (*Risse et al., 2013*). The arena coated with a 2 mm thick layer of 0.8% agar was placed within a temperature controlled LMS220 chamber in the dark at 22°C or at 33 ± 1°C (changes in temperature for short periods of time were due to the inertia of the system). At 33°C the UAS-*shi*[ts] is activated and the synaptic transition is blocked as controlled with *elav*-GAL4/UAS-*shi*[ts] which were paralysed (*Figure 3*). During all treatments at both temperatures the incubator fan was disconnected during the recording to avoid air currents and the larvae were illuminated with infrared light, which they cannot see (HP HSDL-4230 IR LED, 875 nm; intensity from 14.33 nW/mm$^2$ in the edge to 9.12 nW/mm$^2$ in the center). One hour long movies were recorded at 2fps with a Basler acA2040-180km CMOS camera using Pylon and StreamPix software, mounted with a 16 mm KOWA IJM3sHC.SW VIS-NIR Lens and 825 nm high performance longpass filter (Schneider, IF-093). The resolution was 2048 × 2048 px to obtain forward movement displacements and actual pause-turns that are recorded accurately rather than to include 'flickering' movements associated with peristaltic movements.

The *x,y* coordinates of individual larvae exploring the agar were obtained using the FIM track free software (*Risse et al., 2013*). To check that this rate of frame capture was not too slow to accurately and reliably represent a larva trajectory we conducted a larva tracking test at 15 fps, which we then subsampled to 7.5, 5 and 3 fps, to enable a comparison for the same set of tracks at different frame recording frequencies. We found there was no effect of frame capture rate on estimation of the μ exponent parameter (ANOVA: $F$ = 1.96, p=0.146; *Supplementary file 7*) (for track analysis methodologies see sections *Maximum Likelihood Estimation* and *Computing step lengths from tracks*). Therefore, all larva tracks were recorded at two fps. Prior to further analysis, the *x, y* integer coordinates obtained from the tracking software were converted to mm by multiplying by 0.11 (the approximate dimension of 1 pixel).

## Track processing

The processing of the raw tracks from video recordings, required for determining move step lengths, and the subsequent MLE analysis, are described in detail below. In summary however, the workflow process for each raw track proceeded as follows. The track was first smoothed using a Kalman filter, to remove the jaggedness resulting from the integer (pixel) recording. Smoothed tracks were then checked for gap errors, where the time between recorded points exceeded one time step (0.5 s, i.e. missed locations) and for speed errors, where the speed to a point exceeded 0.55 mm/s (recording errors). Turning points in the track were then identified and a set of step lengths was derived. Step lengths below the 0.44 mm minimum step resolution value were discarded and finally the MLE, Mean Square Displacement (MSD) and Straightness Index (SI) analyses were performed.

Each larva trajectory was smoothed with a Kalman filter (*Jonsen et al., 2005*) prior to determining move step lengths to reduce the potential effect of tracking positional errors, e.g. head motion 'flickering', being counted as actual movements. A simple linear smoothing was applied, the aim being to remove the jaggedness of the track caused by the integer (pixel) based recording. To achieve this two 1D filters were used, one for movement in *x* and one for movement in *y*, with parameters for both as follows: position state minimum variance = 0.5; velocity state minimum variance = 0.5; measurement covariance = 1.0. These parameters smoothed the track while introducing a minimum change to the track locations, as shown in *Figure 2A* and *Figure 2—figure supplement 2*. We calculated that any move step <0.44 mm (four pixels) (termed the minimum step resolution value) might not have resulted from actual movement of the larvae but would more likely be the result of noise, or flicker, in the imaging equipment. Therefore, to further reduce the effect of tracking errors, all move steps with values < 0.44 mm were excluded from the analysis. Where tracking errors resulted in gaps in the tracks, the resultant erroneous steps were ignored, as were all steps where a maximum speed of 0.55 mm/s was exceeded. Tracks with fewer than 200 move steps were excluded from the analysis.

## Maximum Likelihood Estimation (MLE) analysis

While there are alternative methods to Maximum Likelihood Estimation (MLE) for the detection of power-law distributions of move step-lengths, such as root mean square fluctuation, or mean squared displacement, these methods do not provide estimates of exponents and cannot be used to properly compare alternate distributions. The methodology used here to determine the best fit distributions is therefore based on that described in *Clauset et al. (2009)* and used previously in Humphries et al. (*Humphries et al., 2012*; *Humphries et al., 2010*). In this study, truncated Pareto (truncated power law, TP) and exponential distributions (E) were the principal models used to test for the presence of Lévy-like walks in move-step frequency distributions. Briefly, the appropriate MLE equation was used to derive an exponent with the initial $x_{min}$ parameter set to the minimum value found in our dataset. A best fit dataset was generated with the estimated parameters and a Kolmogorov-Smirnov (KS) test was used to determine the goodness of fit (the KS D statistic). To determine the best fit value for the $x_{min}$ parameter the calculation was repeated with increasing values for $x_{min}$ taken from the dataset with the value that resulted in the best (lowest) KS-D statistic being retained as the best fit value. The method was repeated to derive a best fit value for the $x_{max}$ parameter, so both the $x_{min}$ and $x_{max}$ parameters were fitted in the same way. In order to provide consistent data sets for model selection, once values for $x_{min}$ and $x_{max}$ had been derived, the dataset was reduced to include only values between those lower and upper bounds. The resulting dataset therefore contained only the step lengths fitting the proposed distribution and it was this that was used to assess the goodness of fit and in the determination of log-likelihood values and Akaike weights, for all distributions, thus simplifying model selection.

The MLE analysis requires two equations for each distribution to be tested. One is the MLE equation for the distribution and is used to estimate the exponent. The other is a random number generator (RNG) and is used to generate best-fit datasets. Note however that there is no closed form MLE equation for the truncated Pareto and therefore we perform the MLE numerically by iteratively searching for the value of μ that maximises the log likelihood. The equations that were used for each distribution are given below. The exponential equations were obtained from *Clauset et al. (2009)*, the truncated Pareto LLH equation was from *White et al. (2008)* and the truncated Pareto RNG was

from Kagan (*Kagan, 2002*). This is the methodology previously described and used in Humphries et al. (*Humphries et al., 2012*; *Humphries et al., 2010*).

*Equation 1*: Exponential MLE

$$\hat{\lambda} = 1 + n\left(\sum_{i=1}^{n}(x_i - x_{min})\right)^{-1}$$

(1)

*Equation 2*: Exponential RNG
Where *r* is a uniform random number on the interval [0, 1]

$$x = x_{min} - \frac{1}{\lambda}\text{In}(1 - r)$$

(2)

*Equation 3*: Truncated Pareto RNG
Where *r* is a uniform random number on the interval [0, 1]

$$x = x_{\min}\left\{r\left[1 - \left(\frac{x_{\max}}{x_{\min}}\right)^{1-\mu}\right] + \left(\frac{x_{\max}}{x_{\min}}\right)^{1-\mu}\right\}^{1/(1-\mu)}$$

(3)

We also compared the truncated Pareto (power-law) distribution with the truncated exponential, log-normal (a heavy-tailed distribution), and gamma distributions (*Humphries et al., 2012*).

## Model selection

To use Akaike Information Criteria weights (*w*AIC), or Akaike weights, as an objective model selection method it is necessary to compute log-likelihoods (LLH) for both of the competing distributions being compared (*Burnham and Anderson, 2004Plank and Codling, 2009*). In this study the principal competing distributions fitted to data were the exponential (e) and the truncated Pareto (TP). The LLH equations for these distributions, respectively, are given as:

*Equation 4*: Exponential LLH

$$LLHe = n(\text{In}(\lambda) + \lambda \cdot x_{\min}) - (\lambda \cdot \sum x))$$

(4)

*Equation 5*: Truncated Pareto LLH

$$LLH_{TP} - n\text{In}\left(\frac{\mu - 1}{(x_{\min}^{1-\mu}) - (x_{\max}^{1-\mu})}\right) - \mu \cdot \sum \text{In}x$$

(5)

All distributions were fitted to the same dataset in each case and therefore the LLH values are directly comparable.

## Computing step lengths from tracks

Various methods have been proposed for the identification of turning points, and consequently move step-lengths, in tortuous two dimensional (2D) movement paths; for example the location of acute turning angles (*Reynolds et al., 2007*), or the deviation of the movement path from an imaginary corridor of set width encompassing the trajectory (*Turchin, 1998*; *de Knegt et al., 2007*). However, the results have been shown to be dependent on the parameters chosen (*Humphries et al., 2013*; *Plank and Codling, 2009*), and it can be difficult to set a threshold turning angle, or a corridor width, that have a sound basis in the biology of the animal and which are not to some degree subjective. These problems were addressed by *Humphries et al. (2013)* by recognising, firstly, that these issues do not arise when dealing with 1D data where the turning points are simply unambiguous reversals in direction and, secondly, that when projected into 1D, the properties of a 2D (or 3D) Lévy distribution are preserved; that is, the 1D projection of a 2 or 3D Lévy walk is itself a Lévy walk. The preservation of the power-law exponent under projection has long been understood and has been described in detail (*Sims et al., 2008*; *Humphries et al., 2010*). Furthermore, other distributions, such as the exponential, were also shown in *Humphries et al. (2013)* to be preserved sufficiently well when projected to be clearly identified as such. Additionally, it was shown that the points at which 1D reversals occurred represented significant turns in the movement path and that

the turn angles at these locations are more uniformly distributed than the turn angles in the raw data, and thus more consistent with a uniform turn angle distribution expected in a pure, idealised Lévy random walk (*Bartumeus et al., 2005*).

Therefore, the method employed here was that described by *Humphries et al. (2013)* which, briefly, consists of taking coordinate values from one axis and computing turning points as the reversals in direction in that axis. Contiguous steps between reversals were considered to be sampling artefacts and were coalesced into single move steps. The process was repeated using the other axis and all steps from both axes were combined into a single move step-length data set for each individual. This method has now been employed in a wide range of movement studies with taxa ranging from bacteria to seabirds (*Reynolds, 2018*) and, importantly, has been proved to work effectively with curved trajectories (*Tromer et al., 2015*).

## Track tortuosity analysis

When plotted at a specific scale, the track of a larva shows little of the underlying complexity, whereas zooming in across different scales reveals complex movement patterns (e.g. *Figure 2B*). To investigate the actual tortuous nature of animal tracks First Passage Time (FPT) analysis can be used (*Bradshaw et al., 2007*). This method computes the variance in the passage forward and backwards of a circle of radius *r* as it is advanced along the track. By increasing the radius and plotting the variance, the scale(s) at which the tortuosity of the track is at a maximum can be determined. This method has been used extensively for the analysis of animal movement tracks, such as albatrosses (*Weimerskirch et al., 2007*). However, while this reveals potential domains or scale(s) of activity it does not provide a time-dependent visualisation of track complexity. To show changes in track complexity at fine spatial scales through time we employed a straightness index (S.I.) that was computed by advancing a window of fixed time scale (e.g. 1 min) along the track point by point and calculating at each point the ratio of the straight line distance to the actual distance travelled. This index, which varies from 0 (no movement) to 1 (straight line movement), can capture very well the complexity of the movement at various scales (set by the window time frame) throughout the track. As the time window decreases the smoothing effect is also reduced revealing increasing track details. A window size of 1 min was selected to balance the need to visualise track complexity without losing the time-series with details of the smallest scale movements. The S.I. was computed for each larva through time and revealed changes in movement as larva alternated between straight line relocation, changes of direction and different degrees of tortuosity. Hence, a change in the S.I. along a larva's track captures the magnitude of the change in movement pattern from intensive, area-restricted searching movements (higher tortuosity) to extensive, straighter line movements (lower tortuosity), and *vice versa*, across a wide range of spatial scales. We analysed the size distribution of S.I. changes (or walk clusters) along each track time-series by fitting model distributions in the same way as with move step-length data, with MLE methods used for parameter fitting and AIC weights (*w*AIC) for model selection.

## Cumulative probability distributions

We used cumulative distribution functions (CDF) to determine the probability of larva track $\mu$ exponents of each experimental treatment (the response) falling within different $\mu$ value ranges. For each of the control treatments *shi*^*ts*^/+ and BLsens > *rpr, hid* at 22˚C, two extreme exponents with values > 2 standard deviations of the mean were discarded prior to analysis. Normality of within treatment log-transformed $\mu$ exponents were examined with Kolmogorov-Smirnov tests and treatment $\mu$ distributions did not differ significantly from normal in each case (all tests resulted in p>0.05) except the control treatment *shi*^*ts*^/+ at 33˚C which was not considered further in this analysis. Using log-transformed individual track $\mu$ exponents we calculated from the CDF the probability of a randomly chosen larva track having a $\mu$ exponent value occurring between 1 and 3 (the Lévy range), 1.25–2.75, 1.5–2.5, or 1.75–2.25 (bounding the theoretically optimum of $\mu$ = 2). All calculations were performed in Minitab 18 (Minitab Inc, Pennsylvania, USA).

## Mean squared displacement

The mean squared displacement (MSD) of each larva trajectory was calculated following *Bartumeus et al. (2005)*. Briefly, for each location in the track the squared straight line distance to

the origin is computed and summed, such that $msd = msd + ((\Delta x_i * \Delta x_i) + (\Delta y_i * \Delta y_i))$ and is plotted as $\log_{10}$(step number) versus $\log_{10}$(msd/step number).

## Acknowledgements

We thank Benjamin Risse and Christian Klämbt for assistance with the building of the experimental set-up, Matthias Landgraf for his support of the research, Lidia Szczupak for the loan and setting up of the electrophysiology equipment, Emiliano Merlo for useful comments on the manuscript, Imogen Sims for help with *Figure 1* and Philip R Braica for Kalman filter code. Funding for track analysis including novel method development was provided by the UK Natural Environment Research Council (NERC) through the Oceans 2025 Strategic Research Programme in which DWS was a principal investigator. D.W.S. was supported by a Marine Biological Association Senior Research Fellowship, VM by a CONICET (PICT 20121578), JB by a Sir Henry Dale Fellowship (Royal Society and Wellcome Trust) Grant 105568/Z/14/Z.

## Additional information

### Funding

| Funder | Grant reference number | Author |
| --- | --- | --- |
| Wellcome | 105568/Z/14/Z | Jimena Berni |
| Royal Society | 105568/Z/14/Z | Jimena Berni |
| Consejo Nacional de Investigaciones Científicas y Técnicas | PICT 20121578 | Violeta Medan |
| Natural Environment Research Council | Oceans 2025 Strategic Research Programme | David W Sims |
| Marine Biological Association | Senior Research Fellowship | David W Sims |

The funders had no role in study design, data collection and interpretation, or the decision to submit the work for publication.

### Author contributions

David W Sims, Conceptualization, Resources, Formal analysis, Supervision, Funding acquisition, Visualization, Methodology, Writing—original draft, Project administration; Nicolas E Humphries, Conceptualization, Data curation, Software, Formal analysis, Visualization, Methodology, Writing—review and editing; Nan Hu, Investigation; Violeta Medan, Investigation, Methodology; Jimena Berni, Conceptualization, Resources, Formal analysis, Supervision, Funding acquisition, Investigation, Visualization, Methodology, Project administration, Writing—review and editing

### Author ORCIDs

Nicolas E Humphries (iD) https://orcid.org/0000-0003-3741-1594
Jimena Berni (iD) https://orcid.org/0000-0002-5068-1372

### Decision letter and Author response

Decision letter https://doi.org/10.7554/eLife.50316.sa1
Author response https://doi.org/10.7554/eLife.50316.sa2

## Additional files

### Supplementary files

• Supplementary file 1. Summary of larvae tracked and the model fits. The number of trials and number of larvae tracked per trial is given along with the move-step frequency distribution model fitting and model selection results for all trials across experimental treatments. The number of larvae paths in TP denotes the number of individual path best fits to a truncated Pareto (power law) distribution; E, the number best fitting an exponential distribution; and U is unclassified where there was no clear

best fit to either model. Tracks discarded prior to model fitting were those where larvae collided, the arena edge was encountered, or there were <50 fitted movement steps. Maximum Likelihood Estimation (MLE) was used for parameter fitting (exponent, $x_{max}$) and Akaike's Information Criteria weights (wAIC) used for model selection. For full description of procedures used see Methods.

• Supplementary file 2. Sensitivity analyses for Kalman filter parameters and minimum step resolution values.(A) Kalman filter (KF) parameters and (B) minimum step resolution values were altered to determine the effects of such changes on the consistency of treatment µ values. The values used for the results presented in the main paper were Position and Velocity minimum variances of 0.5 and a covariance of 1.0 for the KF, and a minimum step resolution of 0.44. The sensitivity analysis for the KF parameters considered values that differed significantly from those used, bracketing the analysis values. As can be seen in (A), differences in the µ values were generally small, confirming that the values chosen for the KF did not alter the finding of truncated power-laws in larva tracks. Rather, the average µ values from all KF sensitivity tests are close to those found in the analysis. The minimum step resolution value chosen for the analysis (0.44) was determined from the tracking resolution and larval movements (head sways and peristaltic contractions) and represents the lowest value above the track noise. All computed move steps lower than this value were excluded from the analysis. For the sensitivity test, values of 0.3, 0.5, 0.7 and 0.9 were used as this range covered viable alternative values. As with the KF tests, the finding of truncated power-laws and the resultant µ values differed very little from those presented in the original analysis. We conclude that significant changes in parameters associated with video track processing had no important effects on our finding of truncated power-laws in larva movement paths and the resultant µ values.

• Supplementary file 3. Summary of results for truncated Pareto model fits compared to exponential model distributions.Full MLE results of truncated power law fits to larvae move step-length frequency distributions across trials and experimental treatments.

• Supplementary file 4. Summary of results for truncated Pareto model fits compared to other model distributions.Model selection using Akaike's Information Criteria weights (wAIC) from comparison of the log likelihoods (LLH). For full description of procedures see Materials and methods. TP, truncated Pareto distribution fit to larva move-step frequency distribution; P, power law; E, exponential; TE, truncated exponential; LN, log-normal; G, gamma distribution. Values in model columns denote best fit based on wAIC. Note the high number of larva best fit by TP model distributions when compared with other models.

• Supplementary file 5. Tests for stationarity in the larva movement pattern data within treatments. Larva tracks with fitted move steps were separated at the midpoint and those for which each half was best fitted by a truncated power-law were retained for analysis. The average µ values for the first and second half of all tracks across trials within a treatment were compared to determine any significant differences, which would indicate changes in track statistics over time (i.e. non-stationarity). We found no significant differences between the first and second halves of the tracks and with no clear trend of increasing or decreasing µ values, as might be expected to occur if µ showed significant temporal dependence on changing satiety or other factors over the 1 hr trial period.

• Supplementary file 6. Summary of model comparisons.Model comparisons for each larva path across trials and experimental treatments is given (wAIC) for the truncated Pareto (TP), exponential (Exp), power-law (P) and composite Brownian (CB) model distributions with proportions of two (CB2), three (CB3) and four exponentials (CB4). Bold wAIC values denote best fit. Note the high number of larva best fit by TP model distributions when compared with other models.

• Supplementary file 7. Video frame frequency test.Ten $shi^{ts}$/+ larvae at 33°C were tracked with a video frame capture rate set at 15 frames per second (fps; Hz). These data were then subsampled to 7.5, 5 and 3 Hz to test whether video frame frequency affected estimation of the $\mu$ exponent following MLE analysis and model selection. A Kruskal-Wallis test showed no differences between medians of truncated power law $\mu$ exponents across the four different frequencies ($H_{(3)}$=3.9, p=0.271) indicating video frame rate did not contribute significantly to determination of $\mu$ exponent values.

• Transparent reporting form

## Data availability

All data generated and analysed in this study are available in Dryad (http://doi.org/10.5061/dryad.7m0cfxpq0). Results from analysis are also included in the manuscript and supporting files.

The following dataset was generated:

| Author(s) | Year | Dataset title | Dataset URL | Database and Identifier |
|---|---|---|---|---|
| Sims DW, Humphries NE, Hu N, Medan V, Berni J | 2019 | Optimal searching behaviour generated intrinsically by the central pattern generator for locomotion | http://doi.org/10.5061/dryad.7m0cfxpq0 | Dryad Digital Repository, 10.5061/dryad.7m0cfxpq0 |

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
