## [Decision Letter]

**Acceptance summary:**

One of the most important open problems in theoretical ecology and movement ecology in particular is whether or not observed Lévy walk movement patterns (and similar behavior) are caused by intrinsic pattern generators implemented as neuronal networks. The alternative possibility is that such movement patterns are an emergent side-effect of how animals interact with the environment. The former explanation suggests that Lévy walks etc. are adaptive, whereas the latter explanation suggests that the behavior is not evolved. This experimental paper gives what is close to a definitive answer to this longstanding question.

The experiment has been custom designed, the data carefully processed, and the analysis has been undertaken with the required high standards (for example, the authors have taken care to distinguish power laws from truncated power laws). There are several technical results that, together, paint a picture that is as surprising as it is significant. When the experimenters artificially impair brain processing of sensory information, the resulting movement paths resemble Lévy walks, supporting the idea that there is an evolved neurophysiological mechanism that generates paths resembling Lévy walks.

**Decision letter after peer review:**

Thank you for submitting your article "Optimal searching behaviour generated intrinsically by the central pattern generator for locomotion" for consideration by *eLife*. Your article has been reviewed by three peer reviewers, and the evaluation has been overseen by Ronald Calabrese as the Senior and Reviewing Editor. The following individuals involved in review of your submission have agreed to reveal their identity: William S Ryu (Reviewer #1); Adam J Calhoun (Reviewer #2).

The reviewers have discussed the reviews with one another and the Reviewing Editor has drafted this decision to help you prepare a revised submission.

Essential revisions:

There are some concerns and needs for clarifications requiring revision. The full reviewer comments are provided and should be fully addressed, but as a guide to the revision, we emphasize a few points brought out in these reviews.

1) Both reviewer #1 (main concerns – request to check if the statistics change over time (e.g. dependent on satiety?)) and #2 (point 4 – is there any adaptation to the turn rate?) overlap and indicate the need for serious consideration by the authors of stationarity in the data.

2) The additional concerns of reviewer #2 can be characterized as two question each requiring a careful response:

a) Is a random walk model (RWM) appropriate for fly larva foraging?

For RWM, the assumption is that reorientation events are independent and that the "steps" do not have some non-trivial correlations (e.g. curvature with direction of turn, etc.). We ask the authors to show additional statistics about turns and steps in support of a RWM. In addition, inspection of the tracks in the figures makes it look like sometimes larvae are either on a curvy track or turning a lot. Our guess is that this is real, and we think some small figure emphasizing that would be not too difficult to produce. For instance, if the authors simply show that slightly different thresholds in the tracking still results in truncated power laws we would be fine with the analysis. Our main concern would simply be that all these power laws are simply a result of the particular tracking.

b) How sensitive are the results to the segmentation parameters?

The authors are looking at 1D of the 2D data set so they aren't identifying "turns" directly and so can't directly do a sensitivity test for "turn angle." Essentially, they are identifying step lengths which statistically reproduce the same exponent of the power law distribution. We ask for a better description of the segmentation pipeline for step lengths in the main text, to help clarify to readers that the analysis does not identify turns directly, since we feel this might be overlooked by some readers. Moreover, we require some comment on how sensitive these results are regarding any parameters used in this pipeline, including image processing, filtering, data exclusion (*x_min_*), and fitting.

Reviewer #1:

This is nice work.

Stochasticity is a fundamental element of animal behavior and in the context of search is universal across a wide range of species. Recent measurements of animal movements has shown that these probabilistic movement patterns approach a Lévy distribution. Leveraging the neurogenetic power of the fly model, this work addresses important questions. How do animals generate these non-trivial distribution? How are these random number generators implemented biologically? Are they generated through sensory information, sensory processing, higher order computation (extrinsically) or through a mechanism at a lower level (intrinsically)?

I believe technically the experiments are sound and that the data analysis uses appropriate and modern statistical tests required for determining the distribution parameters. I especially appreciate the inclusion of the order of magnitude of the data range, and the transparency of the testing details shown in the supplementary files.

The authors are careful to show important controls and experimentally the tracking experiment looks also to be carefully controlled for environmental inhomogeneities. However there are two parameters that I would like the authors to comment on. In *C. elegans* it has been shown that searching behavior changes scale over time, slowly increasing with a tau measured in tens of minutes. Are the fly search patterns constant in time over the 1 hour experiment? Does this differ between the BL/SOG blocked conditions? Also, it is assumed in *C. elegans* that the change in time is due to a change in satiety levels. At the beginning of the experiment the worm is well fed and so when searching on an agar surface with no food, it looks local with a short time/length scale, but as it gets hungry it extends the searching scale. While the blocked BL/SOG condition was shown to be within a range of normal feeding (Figure 2), I wonder if there might be still be a *systematic* nutritional or a satiety difference that is occurring. These comments are directed at a better understanding of mechanism, and does not affect the author's primary argument for an intrinsic vs. extrinsic characterization. But I think might at the very least deserve some comment in a response or perhaps even a comment in the text.

One part of the MLE analysis I did not fully understand. The authors describe a procedure to reduce the data range (*x*_min_, *x_max_*) for the fitting. Why was this necessary? I would like to better understand the differences in fitting if this data pruning was not done. Perhaps one could gleam this information from the supplementary files, but looking it over it is not clear to me.

Reviewer #2:

In the foraging literature, Lévy walks (or truncated power laws) are of fundamental importance and are seen across the animal kingdom. Where do they come from? In this manuscript, Sims et al. propose that power law movement is generated intrinsically by the motion generating circuitry of *Drosophila* larvae. The fits certainly look like power laws and the authors provide evidence that larvae generate this even in the absence of a synaptic activity in the brain. Still, I have some concerns about the way the data is quantified and interpreted.

1) I would like to see some further quantification of how the analysis pipeline identifies turns. This is fundamental to the entirety of the manuscript but when I look at the examples of turns in Figure 4A I am not convinced that there are not large numbers of spurious turns. Are there thresholds that they can vary to ensure that this result isn't the effect of some spurious parameter?

2) I would also like some more insight into how path curvature relates to power law scaling of random walk step length. Many of these paths certainly *look* curved, and I know other worms such as *C. elegans* do have curved paths. Do *Drosophila* larvae? Is a curved path equivalent to a sequence of frequent turns?

3) Some quantification of turning angle seems like it would be useful as well. Can the authors clarify whether optimal search relies on uncorrelated turn angle? If you have a sequence of anticorrelated turns, you may get a power law alpha ~ 2.0 (for the sake of argument) but the animal would move outward in a ballistic fashion which is very different behaviorally. I am especially wondering about turns between short steps which are (I assume) very small and would look very different than the random walks you get in Viswanathan et al., 1999, or the authors' own Figure 1A, for instance.

4) Is there any adaptation to the turn rate, e.g. if you quantify exponent early vs. late is it the same? Many animals perform an area-restricted search when placed in a new environment, which might affect the overall measured exponent.

5) The authors perform a lot of manipulations to see if the animal will continue to produce Lévy flights and indeed they do (I am impressed they do anything with some of these manipulations). In fact, it seems impossible for them to do anything *except* Lévy flights. Have the authors considered a manipulation (puffing on odorants, for instance) that would induce a Brownian random walk and see if they revert to a Lévy flight or if they indeed continue on in a Lévy flight? Some of the manipulations, such as apoptosis experiment in the brain lobes in Figure 7D, suggest that the VNC (not CPGs per se) are what are driving the power law – but there could still be local sensory processing here. My concern here is that the authors may not be truly getting rid of the sensory processing that could drive the power law.

Reviewer #3:

This study is an excellent paper that should be published as soon as possible.

One of the most important open problems in theoretical ecology and movement ecology in particular is whether or not observed Lévy walk movement patterns (and similar behavior) are caused by intrinsic pattern generators implemented neurophysiologically. The other possibility is that such movement patterns are an emergent side-effect of how animals interact with the environment. The former explanation suggests that Lévy walks etc. are adaptive, whereas the latter explanation suggests that the behavior is not evolved. In this context, this paper gives what is close to a definitive answer to this longstanding question.

This is an experimental study that answers an important theoretical problem. The experiment has been custom designed, the data carefully processed and the analysis has been undertaken with the required high standards (for example, the authors have taken care to distinguish power laws from truncated power laws).

There are a number of technical results that, together, paint a picture that is as surprising as it is significant: when the experimenters artificially impair brain processing of sensory information, the resulting movement paths resemble Lévy walks, strongly supporting the idea that there is an evolved neurophysiological mechanism that generates paths resembling Lévy walks.

---

## [Author Response]

Essential revisions:There are some concerns and needs for clarifications requiring revision. The full reviewer comments are provided and should be fully addressed, but as a guide to the revision, we emphasize a few points brought out in these reviews.1) Both reviewer #1 (main concerns – request to check if the statistics change over time (e.g. dependent on satiety?)) and #2 (point 4 – is there any adaptation to the turn rate?) overlap and indicate the need for serious consideration by the authors of stationarity in the data.

We thank the reviewers for raising this important point. We showed in Figure 2—figure supplement 3 that µ values did not change significantly before or after BL/+ and *shi^ts^*/+ control larvae (at 22 or 33^o^C) collided with the edge of the arena, indicating stationarity in the data following intermittence of the path. However, in the revised paper we have undertaken further analysis by taking larva tracks with fitted move steps, separating them at the midpoint, and retaining for analysis those for which each half of the tracks was best fitted by a truncated power law. The average µ values for the first and second half of all tracks across trials within a treatment were compared to determine any significant differences, which would indicate changes in path statistics over time (i.e. non-stationarity). We found no significant differences between the first and second halves of the tracks and with no clear trend of increasing or decreasing µ values, as might be expected to occur if µ showed significant temporal dependence on changing satiety or other factors over the 1 h trial period. We now include these results in a new Supplementary file 5 and refer directly to the results in Figure 2—figure supplements 2 and 3 in the main Results and ‘Exploration strategy in an environment with minimal external cues’ section.

2) The additional concerns of reviewer #2 can be characterized as two question each requiring a careful response:a) Is a random walk model (RWM) appropriate for fly larva foraging?For RWM, the assumption is that reorientation events are independent and that the "steps" do not have some non-trivial correlations (e.g. curvature with direction of turn, etc.). We ask the authors to show additional statistics about turns and steps in support of a RWM. In addition, inspection of the tracks in the figures makes it look like sometimes larvae are either on a curvy track or turning a lot. Our guess is that this is real, and we think some small figure emphasizing that would be not too difficult to produce. For instance, if the authors simply show that slightly different thresholds in the tracking still results in truncated power laws we would be fine with the analysis. Our main concern would simply be that all these power laws are simply a result of the particular tracking.

We thank the reviewers and agree that these are important aspects to clarify further. We address each point in turn below.

Random walks: We agree that it is interesting to consider the appropriateness of the RMW for larva searching. As pointed out, an assumption underlying the concept of a uncorrelated random walk (Brownian motion) is that turn angles follow a uniform random distribution such that any turn angle is equally probable and statistically independent. In the 1980s correlated random walks (CRWs) were developed because CRWs have short-range angle correlations that appear more similar to empirical data since the correlations incorporate directional persistence. However, the memory effects on directional persistence have a finite range and decay exponentially, meaning that at large enough spatial and temporal scales CRWs become uncorrelated random walks. The size of arena we used for *Drosophila* larva is unlikely to allow sufficiently long and large movement paths to emerge such that CRWs emerge as uncorrelated random walks so some directional persistence is expected (non-uniform random turn angle distribution), emerging as an increased frequency of turn angles around 0^o^. This is important to consider in the context of applying Lévy random walks to larva searching paths because an assumption of a pure Lévy walk is that turn angles follow a uniform random distribution. Again though, because our arena size and trial time limited very large spatial and temporal tracks it is expected that movements modelled with Lévy walks will show some short-range correlations.

In the revised paper we have undertaken a turning angle analysis (results given in Figure 5—figure supplement 1) to explore these dynamics. The results demonstrate that using the raw turning angles without applying the objective step identification method results in a highly skewed distribution centred on turns around 0^o^. A raw turn-angle distribution dominated by a very high frequency of small angle deviations around 0^o^ indicate these may be unrelated to actual turning behaviour (interruptions in the movement path). The distribution of significant turns identifies a relatively higher frequency of larger angle turns that is more consistent with a uniform turn angle distribution of idealised random Lévy walks (note that there are far fewer significant turns than raw turns). The significant turns identified between 50 and -50^o^ indicate the presence of short-range angle correlations, however, Lévy walks are independent of short-range correlations and the estimated value of µ_opt_ = 2 remains unchanged under such conditions. This indicates that even with short-range angle correlations the µ values we found for larva will not be affected in important ways.

In the revised paper, therefore, we conclude in the Results subsection ‘Exploration strategy in an environment with minimal external cues’ and in Figure 5—figure supplement 1, that at the scales over which we were able to track larva, the movement statistics are consistent with the expectations of a pure Lévy walk but with short-range angle correlations associated with the spatially and temporally bounded environment.

Curved paths: The reviewers raise an important point here that needed to be clarified further. In the revised paper we have now included a new figure as suggested (Figure 2—figure supplement 1) which explores curved paths and the relationship between curved paths and turning angles.

*Drosophila* larva sometimes exhibit curved paths within movement trajectories, as can be seen in Figure 5 track panels. The new Figure 2—figure supplement 1 shows an example of a Kalman-filtered track of a control larva (BL/+ at 33^o^C) executing a curved path. The figure shows that curved tracks are indeed generated by larvae crawling in a curve which is produced by a consistent but minor deviation in larva heading. We also demonstrate in this figure for the same animal that the objective move-step estimating method (Humphries, Weimerskirch and Sims, 2013; Tromer et al., 2015) identifies small turns within the curved movement. Without more detailed analysis, the path as plotted in this figure gives the impression when plotted at this scale that the method identifies small steps where they appear not to occur. However, a more detailed plot of the actual movements of the larva in the curved track section shows that small turns interrupting the curve are present and are correctly identified by the objective move-step estimation method used.

Threshold analysis: We think it is very important to demonstrate clearly that different thresholds in the tracking still results in truncated power laws. We have undertaken new sensitivity analyses to explore this. The results are presented in a new Supplementary file 2, and referred to directly in the first paragraph of the revised Results.

We altered the Kalman filter (KF) parameters and minimum step resolution values to determine the effects of such changes on the consistency of treatment *µ* values. The values used for the results presented in the main paper were using Position and Velocity minimum variances of 0.5 and a covariance of 1.0 for the KF, and a minimum step resolution of 0.44 mm. The sensitivity analysis for the KF parameters considered values that differed significantly from those used, bracketing the analysis values. The results in Supplementary file 2 show differences in the µ values were generally small, confirming that the values chosen for the KF did not alter the finding of truncated power laws in larva tracks. Rather, the average µ values from all KF sensitivity tests are close to those found in the analysis. The minimum step resolution value chosen for the analysis was determined from the tracking resolution and larval movements (head sways and peristaltic contractions) and represents the lowest value above the track noise. All computed move steps lower than this value were excluded from the analysis. For the sensitivity test, values of 0.3, 0.5, 0.7 and 0.9 were used as this range covered viable alternative values. As with the KF tests, the finding of truncated power laws and the resultant µ values differed very little from those presented in the original analysis. We conclude that significant changes in parameters associated with video track processing had no important effects on our finding of truncated power laws in larva movement paths and the resultant µ values.

b) How sensitive are the results to the segmentation parameters?The authors are looking at 1D of the 2D data set so they aren't identifying "turns" directly and so can't directly do a sensitivity test for "turn angle." Essentially, they are identifying step lengths which statistically reproduce the same exponent of the power law distribution. We ask for a better description of the segmentation pipeline for step lengths in the main text, to help clarify to readers that the analysis does not identify turns directly, since we feel this might be overlooked by some readers. Moreover, we require some comment on how sensitive these results are regarding any parameters used in this pipeline, including image processing, filtering, data exclusion (x_min_), and fitting.

In answer to the question of how sensitive are the results to the segmentation parameters we direct the reviewers to our responses to the previous point. We have substantially altered Kalman filter and minimum step resolution values and found no significant effects on treatment µ values.

However, in the revised paper we have sought to further clarify the description of how larva tracks were processed and turns identified. We have revised the Materials and methods section and included an overview paragraph detailing the process of track analysis. This reads:

“Track processing

The processing of the raw tracks from video recordings, that was required for determining move step lengths, and the subsequent MLE analysis, are described in detail below. […] Turning points in the track were then identified and a set of step lengths was derived. Step lengths below the 0.44 mm minimum step resolution value were discarded and finally the MLE, Mean Square Displacement (MSD) and Straightness Index (SI) analyses were performed.”

We have revised the Materials and methods section referring to the identification of turn angles:

“Computing step lengths from tracks

Various methods have been proposed for the identification of turning points, and consequently move step-lengths, in tortuous two-dimensional (2D) movement paths; for example the location of acute turning angles [Tromer et al., 2015], or the deviation of the movement path from an imaginary corridor of set width encompassing the trajectory [Clauset, Shalizi and Newman, 2009; White, Enquist and Green, 2008]. […] Additionally, it was shown that the points at which 1D reversals occurred represented significant turns in the movement path and that the turn angles at these locations are more uniformly distributed than the turn angles in the raw data, and thus more consistent with a uniform turn angle distribution expected in a pure, idealised Lévy random walk [Bartumeus et al., 2005].”

In the revised paper we have also stated clearly in the first paragraph of the Results section:

“Turns were identified as the reversal in direction in a 1D projection of the 2D movement patterns. This method is unbiased and preserves the distribution properties of the original 2D trajectory [Humphries, Weimerskirch and Sims, 2013] (see Materials and methods).”

Reviewer #1:[…] The authors are careful to show important controls and experimentally the tracking experiment looks also to be carefully controlled for environmental inhomogeneities. However there are two parameters that I would like the authors to comment on. In *C. elegans* it has been shown that searching behavior changes scale over time, slowly increasing with a tau measured in tens of minutes. Are the fly search patterns constant in time over the 1 hour experiment? Does this differ between the BL/SOG blocked conditions? Also, it is assumed in C. elegans that the change in time is due to a change in satiety levels. At the beginning of the experiment the worm is well fed and so when searching on an agar surface with no food, it looks local with a short time/length scale, but as it gets hungry it extends the searching scale. While the blocked BL/SOG condition was shown to be within a range of normal feeding (Figure 2), I wonder if there might be still be a systematic nutritional or a satiety difference that is occurring. These comments are directed at a better understanding of mechanism, and does not affect the author's primary argument for an intrinsic vs. extrinsic characterization. But I think might at the very least deserve some comment in a response or perhaps even a comment in the text.

We refer the reviewer to our responses to point 1 above in which we provide support for stationarity in larva movement data by showing no significant differences in µ values before and after collision, and between the first and second halves of tracks. We now refer to these results explicitly at the beginning of the Results section and in Supplementary file 5 and Figure 2—figure supplement 3.

One part of the MLE analysis I did not fully understand. The authors describe a procedure to reduce the data range (x_min_, x_max_) for the fitting. Why was this necessary? I would like to better understand the differences in fitting if this data pruning was not done. Perhaps one could gleam this information from the supplementary files, but looking it over it is not clear to me.

We do not reduce the data range in order to facilitate the fitting, rather we use the results of the fitting to define the data range for model selection such that across all individual track model fits we are comparing like with like. This is standard methodology in MLE and model selection. However, we have revised the Materials and methods section relating to this (Maximum Likelihood Estimation analysis) to help clarify this point, which now reads:

“In order to provide consistent data sets for model selection once values for *x_min_* and *x_max_* had been derived, the dataset was reduced to include only values between those lower and upper bounds. The resulting dataset therefore contained only the step lengths fitting the proposed distribution and it was this that was used to assess the goodness of fit and in the determination of log-likelihood values and Akaike weights, for all distributions, thus simplifying model selection.”

Reviewer #2:[…]1) I would like to see some further quantification of how the analysis pipeline identifies turns. This is fundamental to the entirety of the manuscript but when I look at the examples of turns in Figure 4A I am not convinced that there are not large numbers of spurious turns. Are there thresholds that they can vary to ensure that this result isn't the effect of some spurious parameter?

We thank the reviewer for helping to identify where we can provide further clarity of our methods and results.

The path tracks shown in Figure 2 and elsewhere all suffer from the same issue that it is difficult to see the fine detail needed to assess that turns are made by the animal itself when the full track is shown at a single, large scale. In Figure 2A (lower panel) it is evident that the animal makes turns at the places where significant turns are identified. However, we understand the need to show more detail about this. In the revision we now show an exploration of the possibility of identifying spurious turns and provide a new analysis (Figure 2—figure supplement 1) demonstrating that small turns are indeed small turns made by the animal and are not an artefact of the methodology.

Furthermore, we would refer the reviewer to our previous responses (point 2A) describing new sensitivity analyses of both the Kalman filter and the minimum step resolution values. We found that broad variations in these parameters have no significant effects on the μ values.

We have revised the Materials and methods to include an overview description of the track processing workflow (see ‘Track processing’ section) and revised the section ‘Computing step lengths from tracks’ to make clear the theoretical basis for the objective method of computing step lengths from identifying significant turns. We also reference clearly in that section the two most relevant papers from different research groups that validate the method (Humphries, Weimerskirch and Sims, 2013; Tromer et al., 2015), including for curved trajectories.

2) I would also like some more insight into how path curvature relates to power law scaling of random walk step length. Many of these paths certainly look curved, and I know other worms such as C. elegans do have curved paths. Do Drosophila larvae? Is a curved path equivalent to a sequence of frequent turns?

We refer to our responses to this point made in the Editor’s Summary (point 2A ‘curved paths’). The *Drosophila* larvae do exhibit curved paths and our new Figure 2—figure supplement 1 shows that curved paths comprise an animal moving in a curve, which is comprised of a sequence of heading deviations to a consistent side of the animal rather than significant turns. (See Turchin 1998 book, for a discussion on deviations and turns.) We find that small turns made by the animal sometimes interrupt this curved heading trajectory, that are unambiguously identified with the objective method employed and which has been shown to work equally well with curved trajectories (Tromer et al., 2015).

3) Some quantification of turning angle seems like it would be useful as well. Can the authors clarify whether optimal search relies on uncorrelated turn angle? If you have a sequence of anticorrelated turns, you may get a power law alpha ~ 2.0 (for the sake of argument) but the animal would move outward in a ballistic fashion which is very different behaviorally. I am especially wondering about turns between short steps which are (I assume) very small and would look very different than the random walks you get in Viswanathan et al., 1999, or the authors' own Figure 1A, for instance.

We refer the reviewer to our response to this point about turning angles in the Editor’s Summary (point 2A ‘random walks’). As mentioned there, we have undertaken a turn angle analysis of truncated power law best fit larva paths and find the expected pattern of turn angles. The distribution is consistent with a near uniform pattern at larger angles and some short-range angle correlations. However, Lévy walks are independent of short-range correlations and the estimated value of µopt = 2 remains unchanged under such conditions. Therefore, optimal Lévy search is not reliant upon an uncorrelated turn angle.

The Results have been revised to include this new analysis, which now reads:

“Furthermore, we also found that the turn angle distribution for all individual larva paths that were best fitted by truncated power laws showed a near uniform distribution of large turn angles (50 and 175°, left or right (Figure 5—figure supplement 1), a characteristic consistent with the uniform turn-angle distribution expected in a pure, idealised Lévy random walk [Viswanathan et al., 1999; Bartumeus et al., 2005]. […] This, taken together with the presence of truncated power law (Lévy) fits to move step-length and walk-cluster size distributions, and the presence of long-range correlations in movement paths confirms the appropriateness of applying the Lévy random walk model to *Drosophila* larva tracks.”

4) Is there any adaptation to the turn rate, e.g. if you quantify exponent early vs. late is it the same? Many animals perform an area-restricted search when placed in a new environment, which might affect the overall measured exponent.

We refer to our response to this point made in the Editor’s Summary (point 1).

5) The authors perform a lot of manipulations to see if the animal will continue to produce Lévy flights and indeed they do (I am impressed they do anything with some of these manipulations). In fact, it seems impossible for them to do anything except Lévy flights. Have the authors considered a manipulation (puffing on odorants, for instance) that would induce a Brownian random walk and see if they revert to a Lévy flight or if they indeed continue on in a Lévy flight? Some of the manipulations, such as apoptosis experiment in the brain lobes in Figure 7D, suggest that the VNC (not CPGs per se) are what are driving the power law – but there could still be local sensory processing here. My concern here is that the authors may not be truly getting rid of the sensory processing that could drive the power law.

The animals do not always perform Lévy walks since in all trials there are some tracks that are either definitely exponential or which cannot be correctly classified as either truncated power law (Pareto) or exponential distributions by the MLE analysis. Supplementary file 2 gives a full summary of the number of tracks fitted by truncated power law, exponential or as unclassified model fits. The movement patterns shown by larva are diverse but there is a predominance of paths best described by truncated Lévy power laws even when brain and sensory processing is blocked.